

# Improved identification of primary biological aerosol particles using single particle mass spectrometry

**Maria A. Zawadowicz[1], Karl D. Froyd[2,3], Daniel M. Murphy[2], and Daniel J. Cziczo[1,4]**

[1]{Department of Earth, Atmospheric and Planetary Sciences, Massachusetts Institute of Technology, Cambridge, Massachusetts}

[2]{NOAA Chemical Sciences Division, Boulder, Colorado}

[3]{Cooperative Institute for Research in Environmental Sciences, University of Colorado, Boulder, Colorado}

[4]{Department of Civil and Environmental Engineering, Massachusetts Institute of Technology, Cambridge, MA, United States}

Correspondence to: D. J. Cziczo (djcziczo@mit.edu)

## Abstract

Measurements of primary biological aerosol particles, especially at altitudes relevant to cloud formation, are scarce. Single particle mass spectrometry (SPMS) has been used to probe aerosol chemical composition from ground and aircraft for over 20 years. Here we develop a method for identifying bioaerosols using SPMS. We show that identification of bioaerosol using SPMS is complicated because phosphorus-bearing mineral dust and phosphorus-rich combustion by-products such as fly ash produce mass spectra with peaks similar to those typically used as markers for bioaerosol. We have developed a methodology to differentiate and identify bioaerosol using machine learning statistical techniques applied to mass spectra of known particle types. This improved method provides far fewer false positives compared to approaches reported in the literature. The new method was then applied to ambient data collected at Storm Peak Laboratory to show that 0.04-0.3% of particles in the 200 – 3000 nm aerodynamic diameter range were identified as bioaerosol.



## 1 Introduction

Primary biological aerosol, hereafter "bioaerosol", include intact and fragmentary microbes, fungal spores and vegetation. One particularly important role of bioaerosol in the atmosphere is that certain species of bacteria and plant material might impact climate via the nucleation of ice in clouds (Hiranuma et al., 2015; Möhler et al., 2008). However, field-based measurements of ice nuclei and ice residuals do not indicate that bioaerosol are a major class of ice active particles (Cziczo et al., 2013; DeMott et al., 2003; Ebert et al., 2011). Uncertainties continue to exist because field measurements of ice nucleating particles are currently sparse. Modeling efforts also suggest that biological material is not significant in ice cloud formation on a global scale. Hoose et al. (2010) has shown that global average contribution of bioaerosol to heterogeneous ice nucleation in mixed phase clouds is small: with higher than realistic freezing efficiencies, the total contribution of biological aerosol remained less than 1%. Later studies by Burrows et al. (2013), Sesartic et al. (2012, 2013) and Spracklen and Heald (2014) support this result. These studies do identify circumstances where bioaerosol can have an influence on clouds. For example, at low altitudes bacteria can dominate immersion freezing rates, where the conditions are too warm for mineral dust to activate (>15°C) (Spracklen and Heald, 2014). Additionally, bioaerosol can dominate the aerosol coarse modes in certain regions. For example bioaerosol can be 50% of the coarse mode over tropical forests compared to a 5-8% global average (Spracklen and Heald, 2014). There are measurements of this enhancement in the Amazon basin, supporting possible regional effects of bioaerosol (Artaxo et al., 1990; Prenni et al., 2009).

Measurement techniques specific to bioaerosol include collection of aerosol on filters followed by analysis with microscopy techniques, either electron microscopy (EM) or optical microscopy coupled with fluorescent staining of the samples (Amato et al., 2005; Bauer et al., 2002, 2008; Bowers et al., 2009, 2011, 2012; Griffin et al., 2001; Matthias-Maser and Jaenicke, 1994; Pósfai et al., 2003a; Sattler et al., 2001; Wiedinmyer et al., 2009; Xia et al., 2013). Aerosol samples collected in the atmosphere have been cultured for identification of the microbial strains present (Amato et al., 2005, 2007; Fahlgren et al., 2010; Fang et al., 2007; Griffin et al., 2001, 2006; Prospero et al., 2005). However, culturing techniques can underestimate microbial diversity, as not all organisms present in the atmosphere are viable or cultivable using standard media. It has been suggested that <10% of bacteria found in atmospheric aerosol are cultivable (Amato et al., 2005; Georgakopoulos et al., 2009).





In-situ techniques specific to biological samples are typically based on fluorescence of
biological material following UV excitation. Examples include the wide-band integrated
bioaerosol sensor (WIBS) which is available commercially (Kaye et al., 2000, 2005). WIBS
has been successfully deployed in several locations (Gabey et al., 2010; O'Connor et al.,
2014; Toprak and Schnaiter, 2013). Using fluorescence to detect biological aerosol can have
interferences, however. For example, polycyclic aromatic compounds or humic acids can
have similar fluorescent properties (Gabey et al., 2010; Pan et al., 1999). Cigarette smoke has
similar fluorescent properties to bacteria (Hill et al., 1999). In an attempt to address
interferences, WIBS collects fluorescence information using several channels with different
wavelengths while also measuring the size and shape of the particles.
Table 1 summarizes recent measurements of bioaerosol in the atmosphere. Apart from WIBS,
the other recent measurements are collection of the aerosol on filters followed by off-line
microscopy. Biological particles have been measured at variety of ground sites, including
urban (Bauer et al., 2008; Fang et al., 2007; Toprak and Schnaiter, 2013), rural (Bowers et al.,
2011; Harrison et al., 2005), forest (Gabey et al., 2010), marine (Griffin et al., 2001; Pósfai et
al., 2003a) and remote (Xia et al., 2013). High-altitude mountain sites, such as Jungfraujoch,
Storm Peak Laboratory, Mt. Rax and Mt. Bachelor Observatory are often used to gain access
to free tropospheric air less impacted by local sources (Bauer et al., 2002; Bowers et al., 2012;
Smith et al., 2012, 2013; Wiedinmyer et al., 2009; Xia et al., 2013). Measured concentrations
range from $2.9 \times 10^3$ to $1.5 \times 10^6$ particles m$^{-3}$, and bioaerosol can make up from 0.5% to 22% of
atmospheric aerosol by number greater than 500 nm. There is a strong seasonal cycle to
biological material (Bowers et al., 2012; Harrison et al., 2005; Toprak and Schnaiter, 2013).
Bioaerosol tend to be primarily bacteria and some fungal spores, although pollen (O'Connor
et al., 2014) and possibly viruses (Griffin et al., 2001) have been reported. Some studies have
performed DNA analysis of bioaerosol, reporting a wide diversity (Smith et al., 2012, 2013;
Xia et al., 2015).
Bioaerosol has also been reported in cloud water (Amato et al., 2005; Bauer et al., 2002;
Sattler et al., 2001) and precipitation samples (Bauer et al., 2002; Christner et al., 2008a,
2008b; Sattler et al., 2001). This does not necessarily mean the bioaerosol play a role in
droplet nucleation processes, however, as scavenging of interstitial aerosol happens frequently
in and below clouds (Pruppacher and Klett, 2003). It does illustrate that microorganisms,





sometimes viable ones, can be transported by the atmosphere and deposited by precipitation
(Amato et al., 2005).
Measurements of bioaerosol in the free and upper troposphere, where they could be relevant
to cloud formation, remain scarce. Four of the recent studies reported in Table 1 used an
aircraft to access altitudes higher than 4,000 m (DeLeon-Rodriguez et al., 2013; Pósfai et al.,
2003a; Twohy et al., 2016; Ziemba et al., 2016). Two of these used the WIBS sensor to report
vertical profiles of fluorescent particles (Twohy et al., 2016; Ziemba et al., 2016). In the
remaining two cases, aerosols were collected on filters and analyzed off-line. Pósfai (2003a)
reported results of Transmission EM (TEM) measurements of samples collected around Cape
Grim that included bacteria with rod-like morphology. It should be noted that numerous other
studies of samples collected on aircraft missions with TEM microscopy did not reveal the
presence of any aerosols that matched morphology of biological material (Buseck and Posfai,
1999; Li et al., 2003a, 2003b; Pósfai et al., 1994, 1995, 2003b). There can exist significant
uncertainty in these measurements. A recent aircraft-based study by DeLeon-Rodriguez et al.
(2013) reports analysis of high altitude (8-15 km) samples taken before, after and during two
major tropical hurricanes. The abundances of microbes, mostly bacteria, were reported
between $3.6 \times 10^4$ and $3.0 \times 10^5$ particles m$^{-3}$ in the 0.25 – 1 μm size range. The methods and
conclusions of this study were re-evaluated by Smith and Griffin (2013), who argued that in
some instances the reported concentration of bioaerosol were not possible because they
exceeded the total aerosol by several factors. The samples were also taken over periods of
hours, possibly including sampling in clouds when the high-speed impaction of droplets and
ice can dislodge particles from the inlet (Cziczo and Froyd, 2014; Froyd et al., 2010; Murphy
et al., 2004).
Although difficult, measurements of bioaerosol in the upper troposphere are necessary in
order to constrain their influence on atmospheric properties and cloud formation processes.
All of the techniques discussed above, except for WIBS, are off-line and require expertise in
sample processing and decontamination. WIBS is a possible in situ detection technique for
bioaerosols, but it is relatively new and, as a result, has a short deployment history. There has
been considerable interest in using aerosol mass spectrometry techniques to measure
bioaerosol. Single particle mass spectrometry (SPMS) has been successfully used since the
mid-1990s to characterize chemical composition of atmospheric aerosol particles in situ and
in real time (Murphy, 2007). The ability of SPMS to simultaneously characterize volatile and



refractory aerosol components makes it an attractive tool for investigating the mechanisms of
cloud formation (Cziczo et al., 2013; Friedman et al., 2013). The general principle behind
SPMS, and in particular the instrument discussed in this paper, the Particle Analysis by Laser
Mass Spectrometry (PALMS), is the use of a pulsed UV laser for the ablation and ionization
of single aerosol particles. Ions are then accelerated into a time-of-flight mass spectrometer.
Laser ablation/ionization used with SPMS produces ion fragments and clusters and is
susceptible to matrix effects such that quantitative results are possible only with careful
calibration and consistent composition (Cziczo et al., 2001).
Biological aerosols have been studied with SPMS, in particular the Aerosol Time of Flight
Mass Spectrometer (ATOFMS; Cahill et al., 2015; Creamean et al., 2013; Fergenson et al.,
2004; Pratt et al., 2009). A property of SPMS bioaerosol spectra that has been exploited for
their detection is the presence of phosphate ($PO^-$, $PO_2^-$, $PO_3^-$) and organic nitrogen ions ($CN^-$,
$CNO^-$) (Cahill et al., 2015; Fergenson et al., 2004). Those ions have also been shown to be
present in non-biological particles with the same instrument, however, such as vehicular
exhaust (Sodeman et al., 2005). One goal of this work is to examine the prevalence of these
ions in the context spectra collected with other SPMSs.
Phosphorus is a limiting nutrient in terrestrial ecosystems (Brahney et al., 2015). On the
global scale, phosphorus-containing dust aerosols are primarily responsible for delivering this
nutrient to oceans and other ecosystems (Mahowald et al., 2008, 2005). Bioaerosols can be an
important source of atmospheric phosphorus on local scales, especially in heavily forested
areas, like the Amazon (Mahowald et al., 2005). The global phosphorus budget has been
modeled by Mahowald et al. (2008), indicating that 82% of the total burden is emitted in the
form of mineral dust. Bioaerosol accounts for 12% and anthropogenic combustion sources,
including fossil fuels, biofuels and biomass burning, account for 5% (Mahowald et al., 2008).
Recently, Wang et al. (2014) provided a higher estimate of phosphorus emissions from
anthropogenic combustion sources, 31%. In this estimate, mineral dust was responsible for
27%, bioaerosol 17% and natural combustion sources 20% of total phosphorus emissions
(Wang et al., 2014). These examples illustrate the major factors in the global phosphorous
budget but also that significant uncertainties exist in the emission inventories. A second goal
of this work is to determine if the non-biological phosphate aerosols, such as those from
minerals and combustion, can be detected and differentiated from bioaerosol.





**2 Experimental**
2.1 PALMS
The objective of this work is to describe and validate a new SPMS-based data analysis
technique that allows for the selective measurement of bioaerosol. A dataset of bioaerosol,
phosphate-rich mineral and coal fly ash single particle spectra – the three largest sources of
phosphorous in atmospheric aerosols - was used to derive a classification algorithm for
biological and non-biological phosphate-containing material. This classifier was then applied
to an ambient data set collected at the Storm Peak Laboratory during the Fifth Ice Nucleation
workshop—phase 3 (FIN03).
The NOAA PALMS instrument has been discussed in detail elsewhere (Cziczo et al., 2006;
Thomson et al., 2000). Currently, there are two copies of the PALMS instrument, both of
which were used in this work. The laboratory PALMS is a prototype for the flight PALMS,
which is more compact and can be deployed unattended at field sites and on aircraft
(Thomson et al., 2000). Briefly, PALMS uses an aerodynamic lens to sample aerosols and
impart them with a size-dependent velocity (Zhang et al., 2002, 2004). Aerodynamic particle
diameter is measured by timing the particles between two continuous-wave laser beams (532
nm Nd:YAG in laboratory PALMS and 405 nm diode in flight PALMS). The particles are
ablated and ionized in one step by a 193 nm excimer laser. A unipolar reflectron time of flight
mass spectrometer is then used to acquire mass spectra. Due to the high laser fluence used for
desorption and ionization ($\sim 10^9$ W/cm$^2$), PALMS spectra show both atomic ions and ion
clusters, which complicate spectral interpretation. SPMS is considered a semi-quantitative
technique because the ion signal depends on the abundance and ionization potential of the
substance, rather than solely its abundance (Murphy, 2007). Additionally, the ion signals can
depend on the overall chemical composition of the particle, known as matrix effects (Murphy,
2007). The lower particle size threshold for PALMS is ~200 nm diameter and is set by the
amount of detectable scattered light. The upper size threshold is set by transmission in the
aerodynamic lens at ~3 μm diameter (Cziczo et al., 2006). The 193 nm excimer laser can
ionize all atmospherically-relevant particles within this size range with little detection bias
(Murphy, 2007). The ionization region is identical in the laboratory and flight PALMS
instruments.
2.2 Test samples



A collection of phosphorus-containing samples of biological and inorganic origin were used
for this work. Some of the samples were analyzed with the laboratory PALMS at the Aerosol
Interaction and Dynamics in the Atmosphere (AIDA) facility at Karlsruhe Institute of
Technology (KIT) during the Fifth International Ice Nucleation Workshop—phase 1 (FIN01)
with the remainder sampled at MIT.
Biological aerosol sampled at AIDA included two aerosolized cultures of *Pseudomonas*
*syringae* bacteria, Snomax (Snomax International, Denver, CO) (irradiated, desiccated and
ground *Pseudomonas syringae*) and hazelnut pollen wash water. The Snowmax and *P.*
*syringae* cultures were suspended in water and aerosolized with a Collison-type atomizer. The
growth medium for *P. syringae* cultures was Pseudomonas Agar Base (CM0559, Oxoid
Microbiology Products, Hampshire, UK).
Biological aerosol sampled at MIT included giant ragweed (*Ambrosia trifida*) pollen, oak
(*Quercus rubra*) pollen, European white birch (*Betula pendula*) pollen, *Fusarium solani*
spores and yeast. Samples of dried pollens and *F. solani* spores were purchased from Greer
(Lenoir, NC). Information supplied by the manufacturer indicates that *F. solani* fungus was
grown on enriched trypticase growth medium and killed with acetone prior to harvesting the
spores. Ragweed and oak pollen originated from wild plants, while the birch pollen originated
from a cultivated plant. Pollen was collected, mechanically sieved and dried. The yeast used
in this experiment was commercial active dry yeast (Star Market brand). The yeast powder
was sampled by PALMS from a vial subjected to slight manual agitation. Pollen grains were
too large (18.9 – 37.9 μm according to manufacturer's specification) to sample with PALMS.
They were suspended in Milli-Q water (18.2 MΩ cm, Millipore, Bedford, MA) and the
suspensions were sonicated in ultrasonic bath for ~30 minutes to break up the grains. Large
material was allowed to settle to the bottom and a few drops of the clear solution from the top
of the suspensions were further dissolved in ~5 mL of Milli-Q water, and the resulting
solutions were aerosolized with a disposable medical nebulizer (Briggs Healthcare,
Waukegan, IL). A diffusion dryer was used to remove condensed phase water prior to
sampling with PALMS. *F. solani* spores were sampled in two different ways: (1) dry and
unprocessed, in the same way as the yeast and (2) fragmented in ultrasonic bath and wet-
generated, in the same way as pollen samples. No processing-related changes to chemistry
were found.





Internally mixed biological/mineral particles were also analyzed at MIT. Illite NX (Clay
Mineral Society) without bioaerosol was sampled dry, using a shaker (Garimella et al., 2014),
and wet-generated, using a medical nebulizer containing Milli-Q water. A second disposable
medical nebulizer was then used to aerosolize a solution of illite NX and *F. solani* spores.
This wet generated aerosol was also dried with a diffusion dryer prior to PALMS sampling.
Samples of fly ash from four coal-fired U.S. power plants were used as proxy for combustion
aerosol: J. Robert Welsh Power Plant (Mount Pleasant, TX), Joppa Power Station (Joppa, IL),
Clifty Creek Power Plant (Madison, IN) and Miami Fort Generating Station (Miami Fort,
OH). The samples were obtained from a commercial fly ash supplier, Fly Ash Direct
(Cincinnati, OH). Fly ash was dry generated with the shaker.
Apatite and Monazite-Ce mineral samples were generated from ~3" pieces of rock. The rocks
were ground and the samples aerosolized with the shaker. Both apatite and monazite were
sampled and processed at MIT. The apatite rock was contributed by Adam Sarafian (Woods
Hole Oceanographic Institution, Woods Hole, MA).
Samples of apatite and J. Robert Welsh Power Plant fly ash were also subjected to processing
with nitric acid to approximate atmospheric aging. Powdered sample was aerosolized from
the shaker to fill a 9 L glass mixing volume. A hot plate below the volume was used to heat
the air inside to 31°C measured in the center of the volume with a thermocouple. PALMS
sampled at a flow of 0.44 slpm from the 9 L volume. This constituted unprocessed aerosol.
80% $HNO_3$ was then placed with a Pasteur pipette at the heated bottom of the mixing volume.
Two experiments were conducted.: for 0.1 mL experiments the entire volume of $HNO_3$
evaporated, producing an estimated partial pressure of about 0.005 atm in a static situation. In
1 mL experiments some liquid $HNO_3$ remained at the bottom of the volume with an estimated
partial pressure of $HNO_3$ of 0.04 atm. The aerosol and gas-phase $HNO_3$ were allowed to
interact for 2 minutes at which point PALMS began sampling from the volume.
Samples of natural soil dust were collected from various locations listed in Table 2. Five
sampled were investigated at the AIDA facility during FIN01 (Bächli soil, Argentina soil,
Ethiopian soil, Moroccan soil and Chinese soil) with the remaining analysis at MIT (Storm
Peak and Saudi Arabian soil). Two samples of German soil were used as an example of
agricultural soil that was known to be fertilized with inorganic phosphate. These were also
sampled at the AIDA facility during FIN01.





2.3 Statistical analysis
A support vector machine (SVM), a supervised machine learning algorithm (Cortes and
Vapnik, 1995), was used as the statistical analysis method for analysis of these data. A portion
of the data from each of the bioaerosol and non-biological phosphate samples was used as
"training data" to build the algorithm. The remaining data were differentiated by the trained
algorithm and the correctness judged based on their source. In this case a non-linear binary
classifier was constructed, using non-linear kernel functions (Ben-Hur et al., 2001; Cortes and
Vapnik, 1995). A Gaussian radial basis function kernel was empirically determined to provide
the best performance in this case. For this work, the SVM algorithm was implemented in
MATLAB 2016a (MathWorks, Natick, MA) using the Statistics and Machine Learning
toolbox.
2.4 Field data
The method was employed on an ambient data set acquired at the Desert Research Institute's
(DRI's) Storm Peak Laboratory located in Steamboat Springs, CO. Storm Peak Laboratory is
located on Mt. Werner at 3220 m elevation at 106.74 W, 40.45 N. This high altitude site is
often in free tropospheric air, mainly during overnight hours, with minimal local sources
(Borys and Wetzel, 1997). Ambient air was sampled using the Storm Peak facility inlet with
the flight PALMS instrument in September, 2015. Measurements were made during Fifth
International Ice Nucleation Workshop—phase 3 (FIN03).
**3  Results**
Figure 1 shows the spectra of biological species: *P. syringae* bacteria, Snomax and hazelnut
pollen wash water particles. These particles contain both organic and inorganic species.
Because they are easy to ionize, the inorganic species sodium and potassium stand out in the
positive spectra despite their minor fraction by mass. Sulfates, phosphates and nitrates are
present, and visible in their associations with potassium. Negative spectra are dominated by
$CN^-$, $CNO^-$, phosphate ($PO_2^-$ and $PO_3^-$) and sulfate ($HSO_4^-$). Higher mass associations of
potassium and sulfates, phosphates and nitrates occur ($K_3H_2SO_3^-$, $K_2H_3NO_4^-$, $K_3H_2PO_2^-$ and
$K_3H_3SO_3^-$). Chlorine is present on some particles. Chlorine is a known contamination from
the Agar growth medium since spectra of aerosolized Agar devoid of bacteria contain large
amounts of chlorine (not shown here).



Figure 2 shows spectra of apatite. In positive polarity, apatite spectra are dominated by
calcium, its oxides, and in associations with phosphate ($CaPO^+$, $CaPO_2^+$, $CaPO_3^+$, $Ca_2PO_3^+$
and $Ca_2PO_4^+$) and fluorine ($CaF^+$, $Ca_2OF^+$ and $Ca_3OF^+$). Negative spectra are dominated by
phosphates ($PO^-$, $PO_2^-$ and $PO_3^-$) and fluorine is often present. Lab-generated apatite spectra
analyzed in this study contain little organic. This may be a result of post-processing of the
apatite sample, in particular the use of ethanol as a grinding lubricant. In contrast, ethanol was
not used in grinding the monazite sample here and its spectra exhibit peaks associated with
organic matter ($C_2H^-$).
Figure 3 shows spectra of coal fly ash from the J. Robert Welsh Power Plant. The positive
spectra contain sodium, aluminum, calcium, iron, strontium, barium and lead. As in apatite,
calcium/oxygen, calcium/phosphate and calcium/fluorine fragments are present. Fly ash
particles also contain sulfate ($H_3SO_3^+$). The negative spectra contain phosphates ($PO_2^-$, $PO_3^-$),
sulfates ($HSO_4^-$) and silicate fragments, such as $(SiO_2)_2^-$, $(SiO_2)_2O^-$, $(SiO_2)_2Si^-$ and $(SiO_2)_3^-$.
The results of $HNO_3$ processing experiments are also shown in Figures 2 and 3. Processing
with nitric acid had an effect on both apatite and fly ash: the calcium/fluorine positive
markers ($CaF^+$, $Ca_2OF^+$ and $Ca_3OF^+$) and the negative fluorine marker ($F^-$) are either reduced
in intensity or completely absent after processing. Additionally, $CN^-$ and $CNO^-$ appear and/or
intensify after processing.
A classifier was designed to use the ratios of phosphate ($PO_2^-$, $PO_3^-$) and organic nitrogen
($CN^-$, $CNO^-$) spectral peaks. This approach has previously been used with PALMS data to
differentiate mineral dusts using silicate and metal peaks to reveal underlying differences in
chemistry (Gallavardin et al., 2008). Figure 4A shows normalized histograms of the $PO_3^-/PO_2^-$
ratio for the test aerosol. The aerosols that contain inorganic phosphorus, such as apatite,
monazite, fly ash and soil dust, cluster at $PO_3^-/PO_2^- < 4$. The bioaerosols cluster at $PO_3^-/PO_2^-$
$> 2$. Processing of apatite with nitric acid tends to shift the $PO_3^-/PO_2^-$ ratio to larger values,
increasing the disparity from the bioaerosols. Ragweed pollen is an exception, with a wide
cluster in $PO_3^-/PO_2^-$ from 1 to 5.
A simple delineation can be made based only on the ratio of phosphate peaks at $PO_3^-/PO_2^- =$
3. The misclassification rate of this simple filter is 20 - 30% for the materials considered here,
with ragweed pollen and fly ash as the greatest sources of confusion between the bioaerosol
and non-biological classes. A lower misclassification between the bioaerosol and non-
biological classes can be achieved if the ratio of organic nitrogen peaks is also taken into





account. Figure 4B shows normalized histograms of $CN^-/CNO^-$ ratios for the test aerosol. In
contrast to $PO_3^-/PO_2^-$ ratios, $CN^-/CNO^-$ ratios do not, by themselves, exhibit a clear difference
between the classes. A superior separation is achieved when data are plotted in a $CN^-/CNO^-$
vs. $PO_3^-/PO_2^-$ space, as shown in Figure 5. In this case two clusters appear. The soil dust class
was left out from the training set because it is not known *a priori* if and how much biological
material it contains (classification with the SVM algorithm is discussed latter). The boundary
between the classes in $CN^-/CNO^-$ vs. $PO_3^-/PO_2^-$ space is non-linear: the SVM algorithm
"draws" this boundary, as shown in Figure 5. The misidentification rate in this 2D
classification is ~3%. As before, ragweed pollen is the cause of most errors; if it is removed
from training dataset, the misidentification rate falls to <1%.
Once trained with the laboratory data, the SVM algorithm was used to analyze the FIN03
field dataset collected at Storm Peak. As a first step, "phosphorus-containing" particles were
identified in the dataset. The criterion for phosphorus-containing used for this work is the
presence of both $PO_2^-$ and $PO_3^-$ ions at fractional peak area (area of peak of interest/total
spectral signal area) greater than 0.01. This threshold was set by examination of the ambient
mass spectra to determine when the phosphate peaks are above the noise threshold. Ambient
particles commonly have small peaks at masses below ~200 due to a diversity of organic
components. The height of this background is ~0.01 and data below this level are considered
uncertain. Phosphorus-containing ambient spectra were then classified by the SVM algorithm
as bioaerosol or inorganic phosphorous if the $CNO^-$ ion was also present at fractional peak
area greater than 0.001. If $CNO^-$ fractional area was less than 0.001, the spectrum was also
classified as inorganic phosphorus.
During the FIN03 campaign, phosphorus-containing particles represented from 0.2 to 0.5% by
number of the total detected in negative ion mode depending on the sampling day and a 0.4%
average for the entire dataset. As shown in Figure 6A when the binary classifier described in
this work was applied to the phosphorus-containing particles, bioaerosol represented a 29%
subset by number (i.e., 0.1% of total analyzed particles). This is within, and towards the lower
end, of previous estimates with biological-specific techniques (Table 1). This lower end
estimate may, in part, be due to PALMS sampling particles in the 200-500 nm diameter range
as well as larger sizes. Previous estimates tend to show increased bioaerosol in the super-
micrometer range and data are often unavailable for the numerous particles smaller than 500
nm diameter.





The origin of the non-biological phosphate particles is likely phosphate-bearing mineral dust
or fly ash. At Storm Peak a likely source is mining of phosphate rock and nearby monazite
deposits. Figure 6B shows HYSPLIT back trajectories for the ten days of the FIN03
campaign; the air masses sampled cross deposits of either phosphate rock (apatite) or rare
earth elements (monazite or carbonatitie). As examples, on 09/27 the back trajectory
intersects the vicinity of an active REE mine in Mountain Pass, CA and on 09/18 and 09/20
the airmass intersected active phosphate mines in Idaho. Although negative spectra of apatite
and monazite cannot be definitively differentiated from fly ash or soil dust spectra, positive
spectra acquired during FIN03 provide additional evidence that monazite-type material was
present. In Figure 2, panels G and H show non-biological phosphate-rich ambient spectra
from FIN03. Figure 2 panels E and F (monazite) contains similar features and matching rare
earth elements.
In total, 56% of phosphate-containing particles analyzed in FIN03 categorized as biological
also contained silicate features. Considered in more detail in the next section, a subset of these
may represent internal mixtures of biological and mineral components.
**4   Discussion**
**4.1 Uncertainty in bioaerosol identification in PALMS spectra**
The method of identification of bioaerosol described here is based on ratios of phosphate and
organic nitrogen peaks. This work is specific to PALMS but can be considered a starting point
from which identification and differentiation can be made with similar instruments. Previous
work with PALMS shows this ratio approach can be used to identify differences in chemistry,
for example among mineral dusts (Gallavardin et al., 2008). In this case the classes are
bioaerosol and non-biological phosphorous; Figure 4A shows that phosphorus ionizes
differently in these classes. In apatite and monazite, phosphorus occurs as calcium phosphate.
In biological particles, phosphorus occurs mostly in phospholipid bilayers and nucleic acids.
In these experiments, the $PO_3^-/PO_2^-$ ratio of those two forms is different (Figure 4A). The
agricultural soils considered here cluster with the minerals and fly ash and we assume the
phosphorous is due to the use of inorganic fertilizer, which is derived from calcium phosphate
(Koppelaar and Weikard, 2013). Fly ash aerosol clusters similarly to apatite and monazite but
with a wider distribution; this is likely because the chemical from of phosphorus in fly ash is
different than in the minerals. Phosphorus present in coal is volatilized and then condenses
into different forms during the combustion process (Wang et al., 2014).





Phosphorus peak ratios in biological particles cluster differently than in inorganic
phosphorous particles with ragweed pollen an exception (Figure 4A). No satisfactory
explanation for this observation has been found although contamination with phosphate
fertilizer cannot be ruled out. The classification error of the biological filter using $PO_3^-/PO_2^-$
and $CN^-/CNO^-$ ratios is 3% with ragweed alone the source of most of the error. This
unexplained behavior is a cause for concern, as the list of biological samples used as a
training set is extensive, but not exhaustive and other exceptions could exist.
During the FIN03 campaign at Storm Peak, 0.2-0.5% of particles by number detected in
negative polarity contained measureable phosphorus (Figure 6A). On most days, the majority
of phosphorus-rich particles were inorganic. Particles with positive spectra showing the
characteristics of monazite coupled to back trajectories over source areas provides evidence of
the origin of the inorganic phosphate particles. Although apatite/monazite particles make up a
small portion of ambient particles at Storm Peak they are potentially interesting not only due
to their possible confusion with biological phosphate but also as a tracer for industrial mining
and processing activities. Currently, such activities are taking place in Idaho and until very
recently at Mountain Pass, CA (U.S. Geological Survey, 2016a, 2016b). Smaller exploration
activities are also taking place at the Bear Lodge, WY and the REE-rich areas in Colorado,
Idaho and Montana are of interest (U.S. Geological Survey, 2016a).
**4.2 Comparison with existing literature**
Previous studies have attempted to identify bioaerosol with SPMS based on the presence of
phosphate and organic nitrate components. Creamean et al. (2013) and Pratt et al. (2009b)
suggested a "Boolean criterion" where the existence of $CN^-$, $CNO^-$ and $PO_3^-$ in a particle
resulted in its classification as biological. If a silicate components were additionally present,
the particle was classified as an internal mixture of mineral dust and biological components
(Creamean et al., 2013; 2014).
The selectivity of this simple three-component filter (presence or absence of $CN^-$, $CNO^-$ and
$PO_3^-$) for biological particles was investigated for PALMS using the test aerosol database with
results shown in Figure 7. The filter successfully picks biological material. However, it also
has a high rate of false positives. For the material that contains inorganic phosphorus (i.e.,
samples known to be devoid of biological material) the three-component filter selects 56% of
fly ash, 56% of agricultural dust and 32% of apatite and monazite. Soil dust is identified as
biological 78% of the time.





The effect of misidentification of inorganic phosphate as biological can be considered in the
context of the atmospheric abundance of the three major phosphate bearing aerosols: mineral
dust, fly ash and bioaerosol (estimates given in Table 3). Because the emissions estimates
vary, the highest fraction of bioaerosol is the case of the highest estimate of bioaerosol
coupled to the lowest estimate of fly ash and mineral dust (Table 3 and Figure 8A).
Conversely, the lowest fraction of bioaerosol is the case of the lowest estimate of bioaerosol
coupled to the highest estimate of fly ash and mineral dust (Table 3 and Figure 8B).
The misidentification rates shown above are then propagated onto the high and low estimates.
As an example, the fraction of aerosol phosphate due to fly ash (1% in the high and 5% in the
low bioaerosol estimate) is multiplied by .56 to indicate the fraction of fly ash that would be
misidentified as biological phosphate with the simple three-component filter. This
misidentification effect is repeated for the mineral dust emission rate and misidentification
fraction. For simplicity, we considered the mineral dust fraction to be desert soils, termed
aridsols and entisols, which are predominantly present in dust-productive regions, such as the
Sahara or the dust bowl (Yang et al., 2013). According to Yang and Post (2011), the organic
phosphate content of those soils is 5-15% but this is a second order effect when compared to
misclassification. In the high bioaerosol scenario 17% of the phosphate aerosol is biological
(Figure 8A) but when misidentification is considered 81% of particles are identified as such
(Figure 8C). In the low bioaerosol scenario 2% of the phosphate aerosol is biological (Figure
8B) but when misidentification is considered 77% of the particles are identified as such
(Figure 8D). This illustrates that simplistic identification can lead to large misclassification
errors of aerosol sources.
Misidentification can also lead to misattribution. Pratt et al. (2009b) analyzed ice residuals
sampled in an orographic cloud and suggested a biological source using the simple three-
component filter applied to spectra containing calcium, sodium, organic carbon, organic
nitrogen and phosphate. The processed apatite spectrum in Figure 2, devoid of biological
material, contains all of these markers. Similar to the Storm Peak dataset, the Pratt et al.
(2009b) wave cloud occurred in west-central Wyoming which is near the Idaho phosphate
rock deposits (Figure 6) and four U.S. states with active mining of phosphate rock for use as
inorganic fertilizer in agriculture (U.S. Geological Survey, 2016b).
The Pratt et al. (2009b) and Creamean et al. (2013, 2014) studies were performed with a
different SPMS, the ATOFMS (Gard et al., 1997; Pratt et al., 2009a). Because the ATOFMS





uses a desorption/ionization laser of a different wavelength (266 nm) the SVM algorithm used
here may not directly translate to that instrument. Instead, the calculation above assumes only
that the misidentification rates between the simple three-component filter and the SVM
algorithm applies.

## 4.3 Soil dust and internal dust/biological mixtures

Soil dust is an important but complicated category of phosphate-containing atmospheric
particles. Modeling studies, such as Mahowald et al. (2008), treat all phosphorus in soil dust
aerosol as inorganic. However, the phosphorus in soil investigated here took both organic and
inorganic forms. Walker and Syers (1976) proposed a conceptual model of transformations of
phosphorus depending on the age of the soil. At the beginning of its development, all soil
phosphorus is bound in its primary mineral form, matching that of the parent material, which
is primarily apatite (Walker and Syers, 1976; Yang and Post, 2011). As the soil ages, the
primary phosphorus is released. Some of it enters the organic reservoir and is utilized by
vegetation, some is adsorbed onto the surface of secondary soil minerals (non-occluded
phosphorus) and then gradually encapsulated by secondary minerals (Fe and Al oxides) into
an occluded form. The total phosphorus content of the soil decreases as the soil ages, due to
leaching. The organic fraction can encompass microorganisms, their metabolic by-products
and other biological matter at various stages of decomposition. Soil microorganisms are the
key players in converting organic phosphorus back into the mineral form (Brookes et al.,
1984). Yang and Post (2011) estimated organic and inorganic phosphorus content of various
soils based on available data. Spodosols (moist forest soils) have the highest fraction of
organic phosphorus (~45%) and aridsols (sandy desert soils) have the lowest (~5%) (Yang
and Post, 2011). Yang et al. (2013) compiled a global map of soil phosphorus distribution and
its forms and found that 20%, on average, of total phosphorus is organic. Wang et al. (2010)
arrive at 34% of soil phosphorus as organic globally.
The biological PALMS filter was applied to several soil dust samples (Table 2) and the
numbers of biological particles in all cases fall within these estimates. As would be expected,
soils collected in areas with less vegetation exhibit smaller biological contributions. We note
that organic phosphorus content is not necessarily a direct indicator of microbes since it also
encompasses decomposed organic matter. At this time, we are not able to delineate between
primary biological and biogenic or simply complex organic (such as humic acids) material.



In the FIN03 field dataset, 56% of particles identified as biological also contained silicate
markers normally associated with mineral dust. This represents and upper limit of particles
that are an internal mixture of dust and biological material. As stated in the last paragraph,
this biological material probably does not consist of whole cells sitting on mineral particles;
such internally mixed mineral dust particle with surface whole or fragments of biological
material are not supported by EM (Peter Buseck, personal communication, 2016). It currently
remains unclear if such internally mixed particles would be counted as biological with an
optical microscope after fluorescent staining.
Internal mixtures of biological and mineral components were generated in the laboratory in
order to investigate this; an exemplary spectrum of such particle is shown in Figure 9. The
spectrum contains alumino-silicate markers consistent with mineral dust together with
phosphate markers that, in this case, come from the biological material. Using the classifier
developed in this paper on the laboratory-generated internally mixed particles correctly
identifies the phosphate signatures to be biological.
**5   Conclusion**
This paper examines criteria that can be used with SPMS instruments to identify bioaerosol.
We propose a new technique of bioaerosol detection and validate it using a database of
phosphorus-bearing spectra. A simple binary classification scheme was optimized using a
SVM algorithm, with a classification error of 3%. Using the binary classifier developed in this
paper, ambient data collected at Storm Peak during the FIN03 campaign was analyzed.
Particles with phosphorus were up to 0.5% by number of all ambient particles in the 200 –
3000 nm size range. On average, 29% of these particles were identified as biological.
Our work expands on previous SPMS sampling that used a more simple Boolean three marker
criterion ($CN^-$, $CNO^-$ and $PO_3^-$) to classify particles as primary biological or not (Creamean et
al., 2013; 2014). We show that the presence of these markers is necessary but not sufficient.
We show a false positive rate of the Boolean filter between 64% and 75% for a realistic
atmospheric mixture of soil dust, fly ash and primary biological particles.
The trained SVM algorithm was also used to measure the biological content of soil dusts.
Different soil dust samples can have different content of biological material with a range from
2 – 32% observed here. Consistent with the literature, samples taken from areas with
vegetation exhibit a higher biological content.



# 1 Acknowledgements

The authors gratefully acknowledge funding from NASA grant # NNX13AO15G, NSF grant
# AGS-1461347, NSF grant # AGS-1339264, and DOE grant # DE-SC0014487. M. A. Z.
acknowledges the support of NASA Earth and Space Science Fellowship. The authors would
like to thank Ottmar Moehler and the KIT AIDA facility staff for hosting the FIN01
workshop and Gannet Hallar, Ian McCubbin and DRI Storm Peak Laboratory for hosting the
FIN03 workshop. The authors thank the entire FIN01 and FIN03 teams for support and Peter
Buseck for useful discussions.





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

| Site | Elevation (m) | Technique | Concentration of bioaerosol detected (particles m$^{-3}$) | % of total particles (size range) | Type of bioaerosol | Reference |
|---|---|---|---|---|---|---|
| **Ground sites** | | | | | | |
| Jungfraujoch | 3,450 | Fluorescent microscopy | $3.4 \times 10^4$ (free troposphere) $7.5 \times 10^4$ (over surface) | NR | Bacteria | Xia et al., 2013 |
| Storm Peak Lab | 3,220 | Fluorescent microscopy | $9.6 \times 10^5 - 6.6 \times 10^6$ | 0.5-5% (0.5-20 μm) | Bacteria (51%) Fungi (45%) Plant material (4%) | Wiedinmyer et al., 2009 |
| Storm Peak Lab | 3,220 | Fluorescent microscopy | $3.9 \times 10^5$ (spring) $4.0 \times 10^4$ (summer) $1.5 \times 10^5$ (fall) $2.7 \times 10^4$ (winter) | 22% (0.5-20 μm) | Bacteria | Bowers et al., 2012 |
| Mt. Rax (Alps) | 1,644 | Fluorescent microscopy | $1.1 \times 10^4$ (bacteria) $3.5 \times 10^2$ (fungi) | NR | Bacteria and fungi | Bauer et al., 2002 |
| Various locations in Colorado | 1,485-2,973 | Fluorescent microscopy | $1.0 \times 10^5 - 2.6 \times 10^6$ | NR | Bacteria | Bowers et al., 2011 |
| Vienna | 150-550 | Fluorescent microscopy | $3.6 \times 10^3 - 2.9 \times 10^4$ | NR | Fungi | Bauer et al., 2008 |
| U.S. Virgin Islands | NR | Fluorescent microscopy | $3.6 \times 10^4 - 5.7 \times 10^5$ | NR | Bacteria and possible viruses | Griffin et al., 2001 |
| Various sites in the U.K. | 50-130 | Fluorescent microscopy | $5.3 \times 10^3 - 1.7 \times 10^4$ (spring) $8.3 \times 10^3 - 1.5 \times 10^4$ (summer) $6.0 \times 10^3 - 1.4 \times 10^4$ (fall) $2.9 \times 10^3 - 1.0 \times 10^4$ (winter) | NR | Bacteria | Harrison et al., 2005 |
| Danum Valley, Malaysian Borneo | 150-1,000 | WIBS | $2.0 \times 10^5$ (above forest canopy) | NR | FBAP | Gabey et al., 2010 |



| | | | 1.5×10$^6$ (below forest canopy) | | | |
|---|---|---|---|---|---|---|
| Karlsruhe, Germany | 112 | WIBS | 2.9×10$^4$ (spring)<br>4.6×10$^4$ (summer)<br>2.9×10$^4$ (fall)<br>1.9×10$^4$ (winter) | 4-11% (0.5-16 μm) | FBAP | Toprak and Schnaiter, 2013 |
| **Aircraft campaigns** | | | | | | |
| Cape Grim | 30-5,400 | TEM | NR | 1% (>0.2 μm) | Bacteria | Pósfai et al., 2003 |
| Flights around the Gulf of Mexico, California and Florida | 3,000-10,000 | Fluorescent microscopy | 3.6×10$^4$ – 3.0×10$^5$ | 3.6-276% (0.25-1 μm)* | Mostly bacteria | DeLeon-Rodriguez et al., 2013 |
| Flights over southeastern U.S. (SEAC$^4$RS) | Vertical profiles up to 12,000 | WIBS | 3.4×10$^5$ (average, <0.5 km)<br>7.0×10$^4$ (average, 3 km)<br>1.8×10$^4$ (average, 6 km) | 5-10% (0.6-5 μm) | FBAP | Ziemba et al., 2016 |
| Flights over Colorado, Wyoming, Nebraska and South Dakota | Vertical profiles up to 10,000 | WIBS | 1.0×10$^4$ – 1.0×10$^5$ (<2.5 km)<br>0 – 3.0×10$^3$ (>2.5 km) | NR | FBAP | Twohy et al., 2016 |



1    Table 2. Soil dust samples used in this work. The last column shows the results of analysis

2    with the biological filter developed here as a percentage of negative particles sampled.

| Sample | Site description | Approx. collection coordinates | % biological |
|---|---|---|---|
| Bächli | Outflow sediment of a glacier in a feldspar-rich granitic environment. No vegetation. | 46.6 N, 8.3 E | 6.0 |
| Morocco | Rock desert with vegetation. Close proximity to a road. | 33.2 N, 2.0 W | 20.4 |
| Ethiopia | Collected in Lake Shala National Park from a region between two lakes. Area vegetated by shrubs and acacia trees. | 7.5 N, 38.7 E | 32.1 |
| Storm Peak Lab | Collected near Storm Peak Lab. Grass and shrubs present. | 40.5 N, 106.7 W | 31.3 |
| Argentina | La Pampa province. Top soil collected from arable land with sandy loam (Steinke et al., 2016). | 37 S, 64 W (approximate) | 21.3 |
| China/Inner Mongolia | Xilingele steppe. Top soil collected from a pasture with loam (Steinke et al., 2016). | 44 N, 117 E (approximate) | 2.0 |
| Saudi Arabia | Various samples from several locations. Arid, sandy soils. | 24.6 N – 26.3 N, 46.1 E – 49.6 E | 14.5 |





1    Table 3. Literature estimates of emission rates of primary biological particles, dust and fly

2    ash.

| Particle | Emissions (Tg yr$^{-1}$) | |
| --- | --- | --- |
| | *low estimate* | *high estimate* |
| Dust | 1490 (Zender, 2003) | 7800 (Jacobson and Streets, 2009) |
| Primary biological | 186 (Mahowald et al., 2008) | 298 (Jacobson and Streets, 2009) |
| Fly ash | 14.9 (Garimella et al., 2016) | 390 (Garimella et al., 2016) |



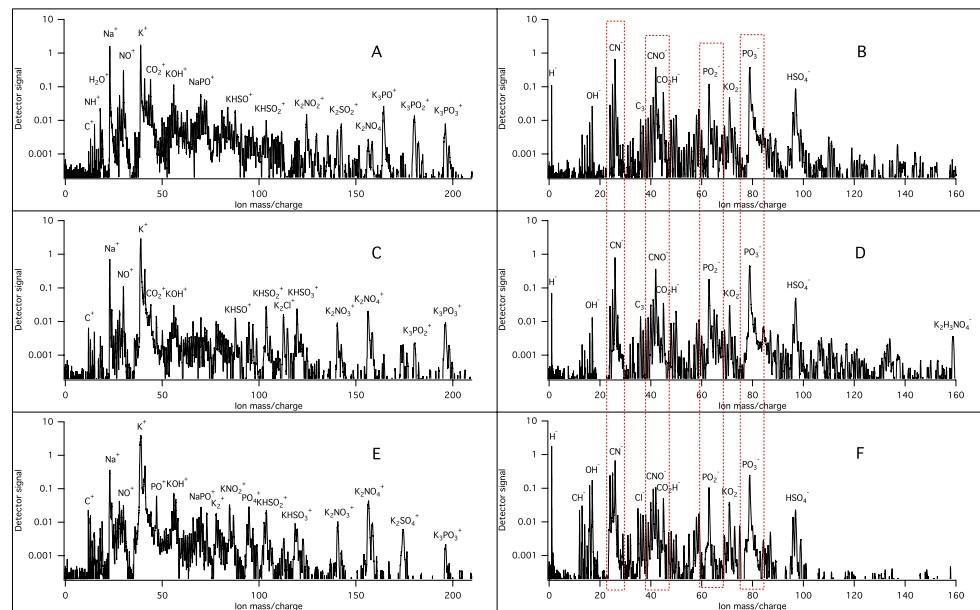

Figure 1. Representative PALMS spectra of bioaerosol. A and B: Snomax. C and D: *P.*
*syringae*. E and F: Hazelnut wash water. Right and left columns are positive and negative
polarity, respectively. Red dotted lines are features indicated in the literature as markers for
biological material.



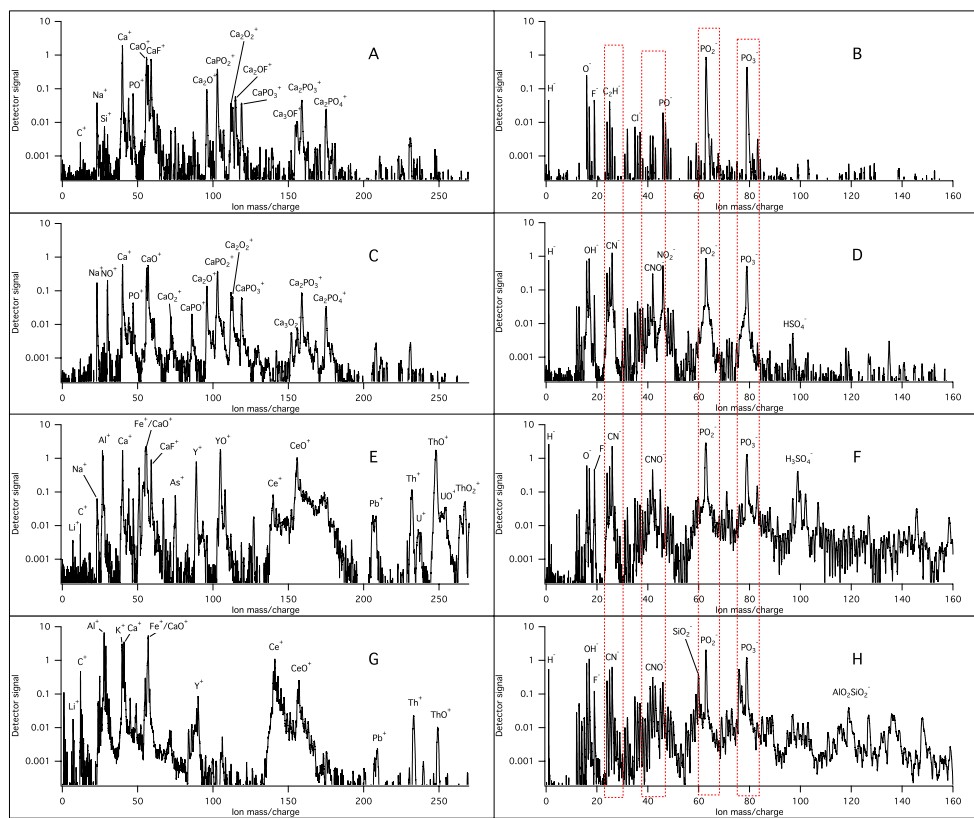

Figure 2. Representative PALMS spectra of phosphorus-rich minerals and ambient aerosol. A
and B: Unprocessed apatite. C and D: Apatite processed with $HNO_3$ (see text for details). E
and F: Monazite-Ce. G and H: Ambient particles sampled at Storm Peak matching monazite
chemistry. Right and left columns are positive and negative polarity, respectively. Red dotted
lines are features indicated in the literature as markers for biological material.





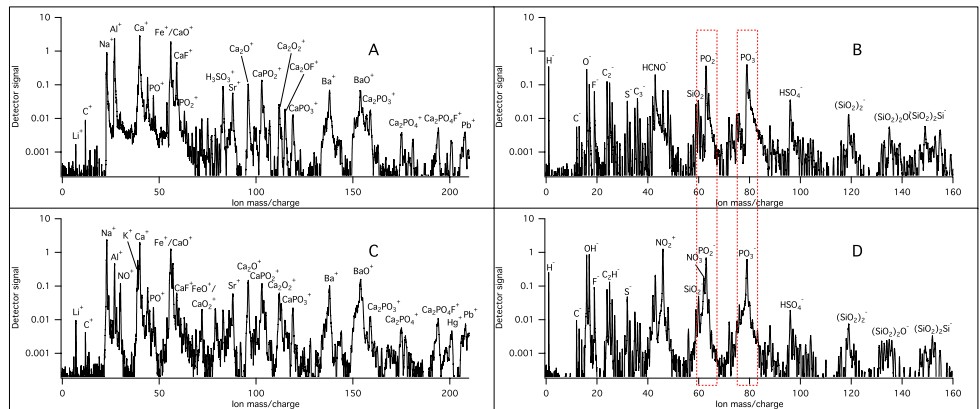

Figure 3. Representative PALMS spectra of coal fly ash from the J. Robert Welsh power
plant. A and B: Unprocessed fly ash. C and D: Fly ash processed with HNO₃ (see text for
details). Right and left columns are positive and negative polarity, respectively. Red dotted
lines are features indicated in the literature as markers for biological material.



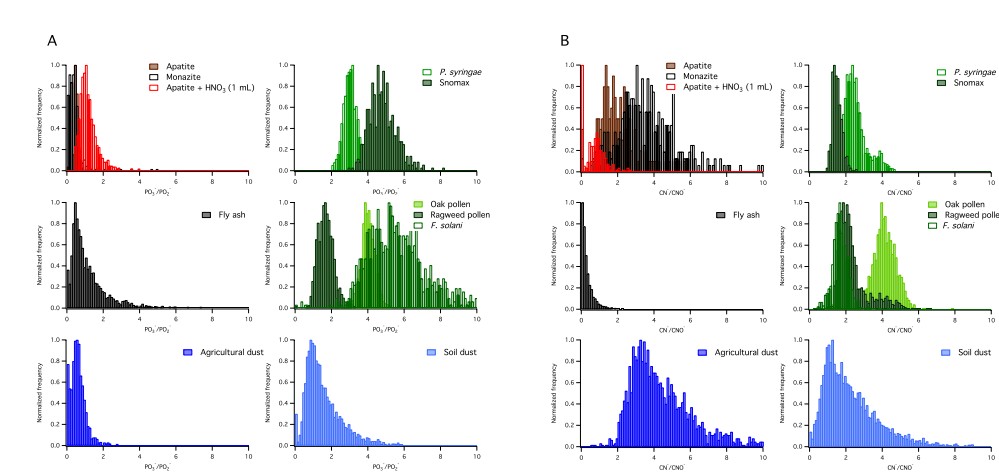

Figure 4. A: Normalized histograms of the $PO_3^-/PO_2^-$ ratio for the test aerosol. B: Normalized
histograms of the $CN^-/CNO^-$ ratio for the same test aerosol as in A. Delineation between the
clusters at a $PO_3^-/PO_2^-$ ratio of 3 results in a 70-80% classification accuracy depending on the
species considered.





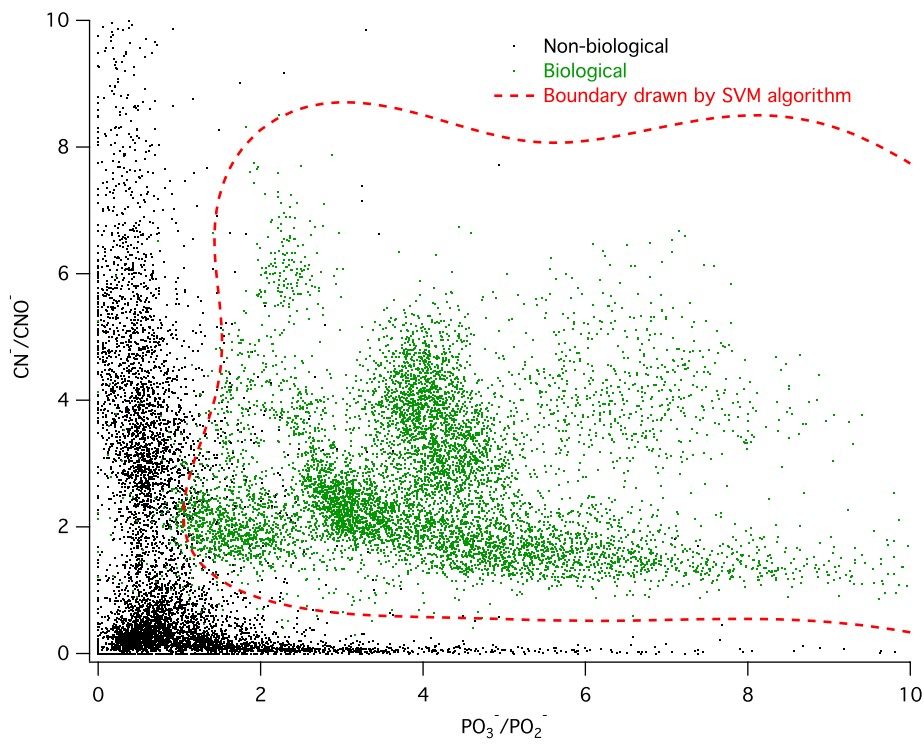

2    Figure 5. Inorganic and biological particle clusters in $CN^-/CNO^-$ vs. $PO_3^-/PO_2^-$ space. The

3    SVM algorithm delineates between the clusters with the red dashed line with an overall 97%

4    classification accuracy.



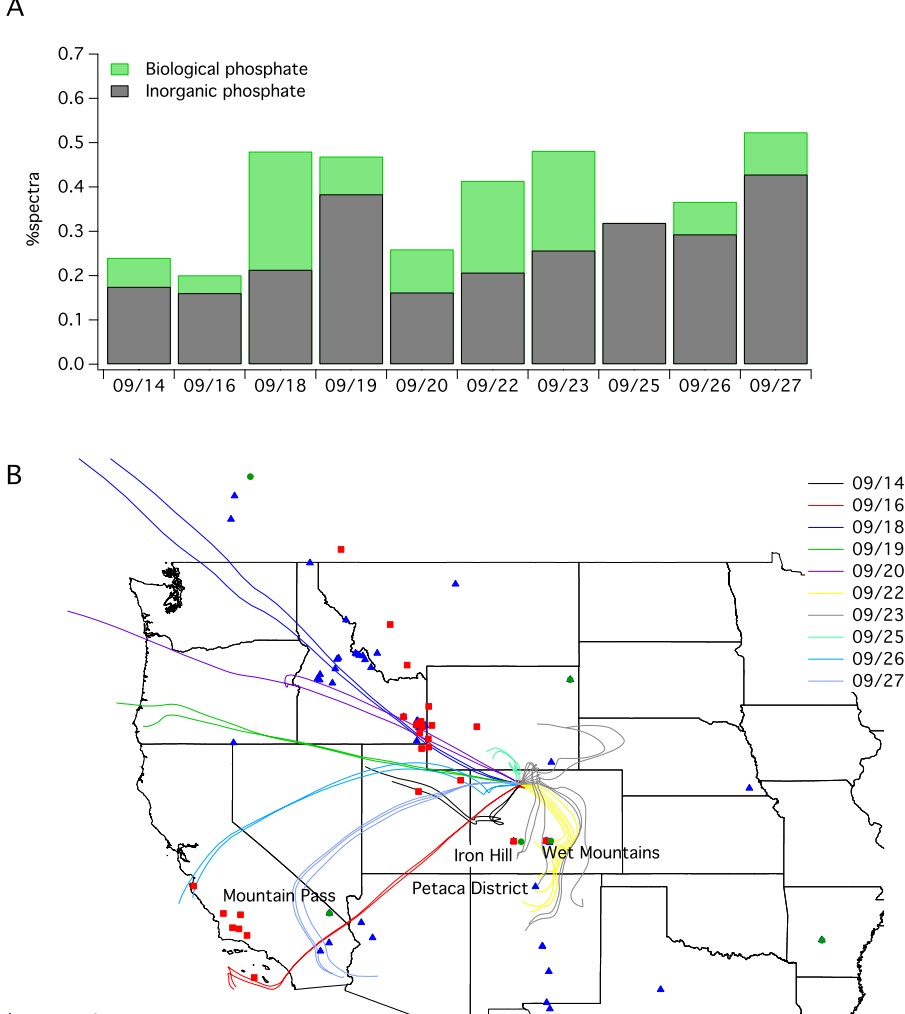

Figure 6. A: The percentage of ambient aerosol particles from the dataset categorized as
biological and inorganic (phosphate-bearing mineral dust or fly ash) phosphate using the
criteria developed in this work. Note that at this location and time of year inorganic phosphate
dominates biological. B: HYSPLIT back trajectories plotted for ten measurement days at
Storm Peak Laboratory. Locations of REE, phosphate and carobonatite deposits, sourced from
U.S. Geological Survey, are co-plotted (Berger et al., 2009; Chernoff and Orris, 2002; Orris
and Grauch, 2002).





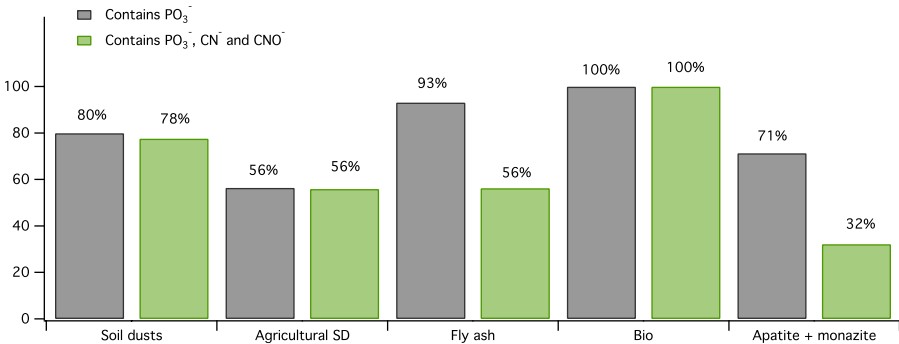

Figure 7. Percentage of particles that include $PO_3^-$, $CN^-$ and $CNO^-$ markers in five classes of
atmospherically-relevant aerosol spectra acquired with PALMS in this work. Note that the
green bars indicate the percentage of particles of each type identified as biological using
literature criteria. In the case of bioaerosol the identification is correct. In all other aerosol
classes the green bar denotes a level of misidentification.





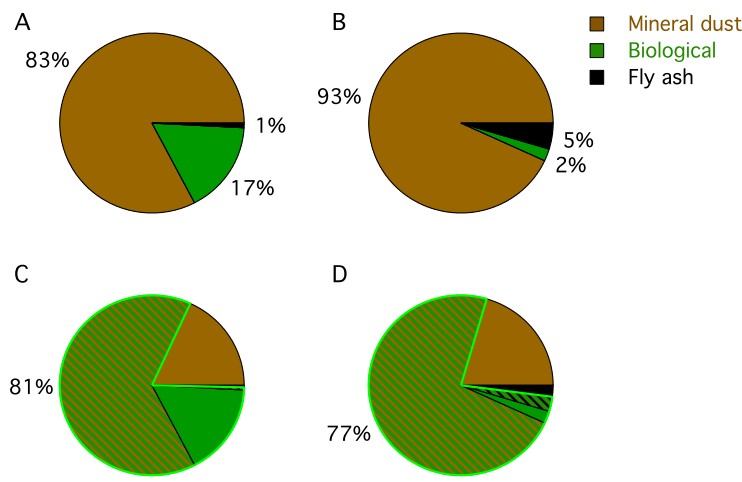

Figure 8. Abundance of bioaerosol, mineral dust and fly ash in the atmosphere constructed
using emissions estimates in Table 3 A: Highest estimate for bioaerosol coupled to lowest
estimates for dust and fly ash. B: Lowest estimate of bioaerosol in the atmosphere coupled to
highest estimates for dust and fly ash. C and D: Effect of misidentification of phosphate- and
organic nitrogen-containing aerosol as biological using the emissions in A and B,
respectively. The hatched regions correspond to the misidentified fractions of mineral dust
and fly ash. In these estimates the correct emissions (solid green region) in A and B (17 and
2%, respectively) are overestimated (hatched green region of misidentified aerosol plus solid
green region) in C and D (as 81 and 77%, respectively).





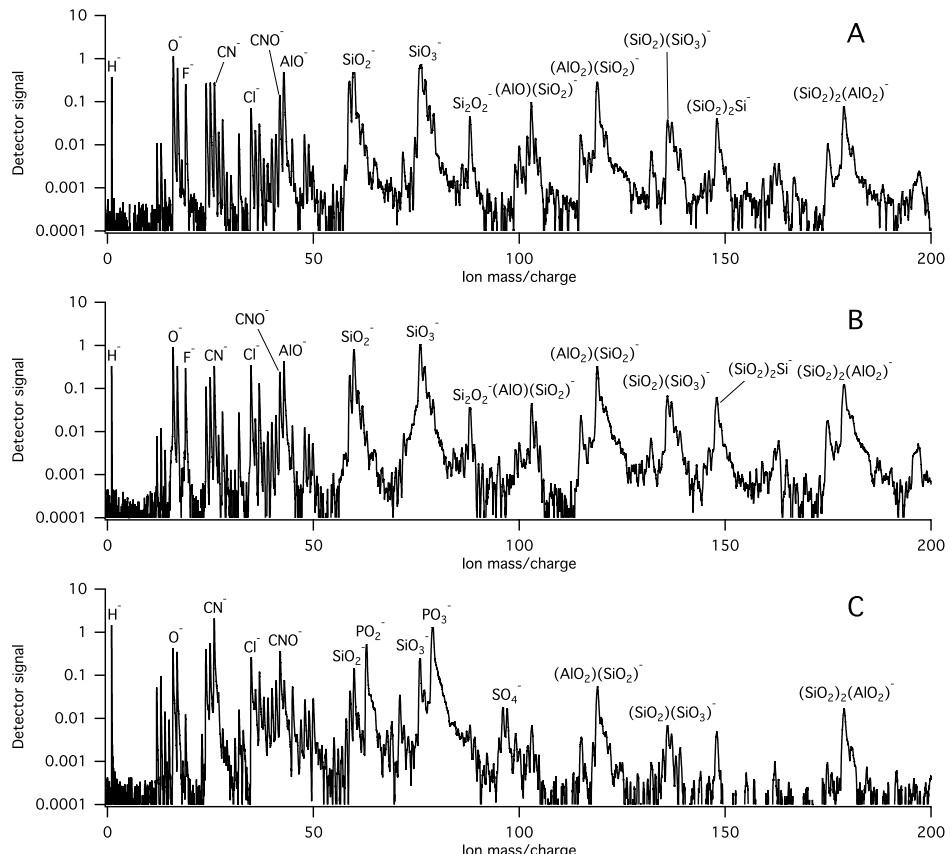

Figure 9. Exemplary PALMS negative polarity spectra of A: dry-dispersed illite NX, B: wet-
dispersed illite NX from a distilled, deionized water slurry and C: similarly wet-dispersed
illite NX but from a water slurry that also contained *F. solani* spores. Note that phosphate
features are absent in A and B but present in C due to addition of biological material to the
mineral dust.