# Peer review of "Improved identification of primary biological aerosol"

_Atmospheric Chemistry and Physics, 2016_

## Referee Comment (RC1) · Anonymous Referee #1 · 13 Jan 2017

In their manuscript "Improved identification of primary biological aerosol particles using single particle mass spectrometry", Zawadowicz et al. present a new classification method to distinguish single particles containing "biological" and "inorganic" phosphorus from their mass spectra. The algorithm is developed and tested using single particle mass spectra acquired in laboratory measurements for a number of various different inorganic (mineral dust, fly ash) and biological (pollen, yeast, fungal spores) test aerosols. The applicability of a previously described method for determining bioaerosol using specific markers is also tested on this sample dataset. The new algorithm is applied to a dataset from ambient measurements to determine the number fraction of biological particles.

[Figure]

The presented method to distinguish biological and inorganic phosphorus-containing particles from single particle mass spectra is clearly improved compared to older methods and in principle should be published. However, the manuscript is focused more on the method development instead of the atmospheric application, which seems only a very minor addition to the paper and doesn't provide much new insight beyond a proof-of-concept. My feeling therefore is that the manuscript would be more suited for a journal more dedicated to the technical aspects of aerosol measurements, like Atmospheric Measurement Techniques.

My main concern however is that while the method itself seems fine, the manuscript is lacking a clear, critical discussion not only of the potentials, but also the limitations of the method. While in the abstract and most of the manuscript the impression is given primary biological aerosol particles (PBAP) could be clearly distinguished from other particles, only in the last section of the discussion (4.3) it becomes apparent that this is not the case at all: in fact, what can be distinguished with the algorithm is whether the particles contain "biological" or "inorganic" phosphorus. Though valuable, this is a very different kind of information than suggested in the title ("identification of primary biological aerosol particles"), the abstract ("identifying bioaerosols", P1, L19, "identification of bioaerosol", P1, L19, "differentiate and identify bioaerosol", P1, L22/23; "in ambient data...0.04-0.3% were identified as bioaerosol", P1, L25-27), and most of the manuscript (several instances, e.g. P11 L20, P11 L26, P16 L16, P16 L17, of misleading use of "bioaerosol", which is defined in P2 L2 as "primary biological aerosol"). Therefore, it needs to be stated much clearer throughout the manuscript what this classification method indeed is capable of, and what its limitations are. Otherwise the false impression is given that PBAP could be detected within ambient aerosol with this method, which clearly is not the case. Consequently, these issues need to be addressed before I could favor publication of this manuscript.

Therefore, I would suggest the authors to re-submit the paper, possibly to a more suitable journal, after performing some major revisions addressing these issues.

Specific comments:

While the paper overall is clearly written, it partially could be more concise and more clearly structured. Especially the introduction is very long, and in parts reads more like a review than like a general introduction to a research paper, esp. P2, L22 to P4, L28. This also leads to an excessively long list of references. Thorough reviews on PBAP and their measurements can already be found e.g. in (Després et al., 2012) and (Georgakopoulos et al., 2009); instead of citing a huge number of individual works, the citation of such review papers might be preferable. By streamlining the introduction especially in P2, L22 to P4, L28 and giving it more focus, the introduction could be significantly shortened while still maintaining all the important information relevant for the manuscript. On the other hand, the discussion on previous efforts in bioaerosol detection using single particle mass spectrometry, which is the most important aspect for the discussions within the manuscript, is very brief and could be expanded (P5 L9-16).

It is not clear to me why parts of the results can be found in the results section, and other parts (like the soil and internal mixtures, Sect. 4.3) in the discussion section. Maybe a single "results and discussion" section would be more appropriate, and could also avoid some repetitions.

Section 4.2: This is an important discussion, and I would have hoped to find a similarly critical discussion of the newly developed algorithm in this manuscript, as well. There are some basic approaches to such a discussion scattered throughout the manuscript, but this should be addressed much more clearly and explicitly. By the title of Section 4.1 it is suggested that this discussion is provided in that section, but in fact the presented discussion on uncertainties and limitations of the newly developed algorithm in that section is very limited and should be much more thorough.

For example:

- In Sect. 3 a "misidentification rate" is given. This needs to be explained in more detail

and more clearly:

1.) What kind of misidentification exactly is contained within this value? Both false positives (mineral dust wrongly assigned to "biological phosphorus") and false negatives (biological material not assigned to "biological phosphorus") cause an uncertainty in the determined fraction of particles containing biological phosphorus.

2.) The method used to calculate the "misidentification rate" should be clearly stated. Depending on the method, the number of particles within the data sets of the different test aerosols might bias the determined uncertainty, so this should be made clear to the reader.

3.) It is stated that removing ragweed pollen from the training set leads to a smaller misidentification rate. Was ragweed pollen only removed from the training set, or also from the "testing" set? I guess the former, since the latter would give a wrong impression, but this needs to be stated clearly.

4.) Which particles were tested for determining this misidentification rate? The pure biological / pure mineral dust / fly ash particles? What about the processed mineral dust?

- It should be discussed in Sect. 3 / Sect. 4.1 what effect has

1) processing of the mineral dust (which, as stated on P10 L17, causes CN- and CNO- to "appear and/or intensify", so might have an influence on the classification), and

2) mixing mineral dust / biological material. This is discussed only in Sect. 4.3, but is an important consideration when assessing the uncertainty and limitations of the method. The discussion in Sect. 4.3 reveals several limitations which need to be discussed within this context: "At this time, we are not able to delineate between primary biological and biogenic or simply complex organic (such as humic acids) material." (P15 L30-31) This means that with the presented method, not PBAP can be determined, but whether phosphorus present in any particle is of "biological" or "inorganic" nature. The former

hints at the presence of biological material, but, as also evident from the discussion on P16 L1-14, it is not possible to determine whether this biological material is part of PBAP or from an internal mixture of e.g. mineral dust and biological material, so the information retrieved remains limited, which needs to be clearly stated and discussed.

- If I understand correctly, only an internal mixture of biological material with a type of mineral dust not showing any signatures of inorganic phosphorus (illite) was tested. But what happens if mineral dust showing mass spectral signatures of inorganic phosphorus (like apatite) is internally mixed with biological material? If indeed in an ambient dataset up to 56% of all particles identified as containing biological phosphorus also contained silicate markers (P16 L1), this does not seem to be an unlikely case and needs to be addressed.

All in all, a clearer discussion of the potentials and limitations of the method is needed: it is capable of differentiating biological and inorganic phosphorus under the tested conditions (within the uncertainty and the limitations to be discussed), but (at least in its present state) it is not capable of distinguishing PBAP. The misleading references to "bioaerosol" throughout the manuscript need to be reworded to reflect this.

2. Experimental section:

In order to get an idea of the underlying statistics, the general information on how many mass spectra (positive/negative) were available for the different samples needs to be included somewhere (in the experimental or the results section), also, how many mass spectra were acquired in the field campaign. If only some of the spectra were used for the analysis, their number (and criteria for their selection) needs to be stated.

How were peak intensities determined for the various ratios (CN-/CNO- etc)? Integrated peak area? This should be stated in the methods section.

P6, L3-9 (first paragraph): this paragraph is not related to the section (2.1: PALMS), but a general introduction. It should go either as a general remark in the experimental

section before Sect. 2.1, or be reworked as a last paragraph into the introduction.

P6, L18: "...a unipolar reflectron time of flight mass spectrometer was used..." – It should be clearly stated here that the PALMS acquires for each single particle either a negative or a positive mass spectrum, but not both simultaneously. How long were the sampling times in positive / negative mass spectra mode (e.g., switching every minute, every 15 min, every hour)?

Could Section 2.2 (Test samples) be streamlined a bit to be more concise? It would be good to have a table with an overview of the sampled materials; maybe some of the detailed information could go into such a table as well, to make the section easier to read and to provide a better overview for the reader.

P7 L25: "...further dissolved in ∼5ml of Milli-Q water..." – this information is not necessary since the concentration of the original solution is not known, anyway.

P7, L30-31: "No processing-related changes to chemistry were found." – This sentence should be clarified, e.g., "...were found in the mass spectra sampled with the PALMS".

P8 L4 "to aerosolize a solution of illite NX and F. solani spores" – I guess this should read "suspension" instead of "solution"? Was this suspension sonicated as well?

P8 L21: "for 0.1 mL experiments" is unclear. Rather something like "For experiments using 0.1 mL of nitric acid"?

P9, L3: "a portion of the data" – how many spectra? Give at least the order of magnitude. The same for the "remaining data" (P9 L5). Also specify which sample types were included for the training: on P11 L5 it is mentioned that soil data were not used. Were all other lab samples used? Please be more specific. Related to this, in Sect. 3, it is stated that for the ambient data, a threshold was used to determine mass spectra containing phosphorus in a first step. Was something similar performed for the lab data (also for the training), or were all mass spectra used?

P9, L18: Please give the start / end dates of the measurement period.

3. Results:

P10 L19 to L27: which bioaerosol materials were used for this? In Fig. 4 only 5 of the 8 tested materials are shown. Were the others used as well, showing similar behavior, and only were omitted for clarity? Please state clearly which materials were used for the development and if some were left out, why. – On P11 L4 it is stated that soil dust is left out from the training set because some biological material might be contained in the particles, however, in P10 L24 it is used within the class of inorganic phosphorus. If biological material is indeed present within soil dust, this does not make sense. This seems like a contradiction.

P10 L26 "Processing of apatite with nitric acid tends to shift the PO3-/PO2- ratio to larger values, increasing the disparity from the bioaerosols." – This is not clear to me. If inorganic material usually shows lower PO3-/PO2- ratios than bioaerosol, shouldn't this read "decreasing the disparity"?

P11 L6: "(classification with the SVM algorithm is discussed later)": maybe this could be reworded for clarity, as at first reading it seems to mean that the SVM algorithm itself will be discussed later, not the results of applying it on the soil samples.

4. Discussion:

P13 L10-12: "Particles with positive spectra showing the characteristics of monazite...provides evidence of the origin of the inorganic phosphate particles." (and a similar statement on P12, L9): Since PALMS does not simultaneously provide the positive and negative ion mass spectrum of a single particle, this is not "evidence", but rather "suggests" this type of mineral dust particles as a likely origin. Please reword.

P13 L26 and following: What thresholds were used in order to determine whether the different marker ion signals were present or absent? Was it tried to improve the performance of the algorithm by adjusting these thresholds?

P15 L26/27: "the numbers of biological particles fall within these estimates". The cited

estimates refer to inorganic / biological phosphorus mass ratio, while the information provided by PALMS is the number ratio. Such a comparison would only be valid if it can be assumed that all types of particles contain the same amount of phosphorus, which does not seem very likely. This needs to be discussed more carefully. A similar comment applies to P16, L28-31.

Figures and tables:

Figs 1-3, Fig. 9: The mass spectra might be easier to grasp if integrated stick spectra were shown instead of the raw mass spectra.

Fig. 7: "In all other aerosol classes the green bar denotes a level of misidentification." – This only applies to "Apatite + Monazite" and "Fly ash". Both "agricultural soil dust" and "Soil dusts" are expected to contain some (unknown) amount of biological material, so the performance of the algorithm cannot be validated on these samples.

Technical corrections:

Various locations, e.g., P6 L6, P12 L23, P12 L28: "phosphorous" should read "phosphorus"

P7 L8: "Snowmax" should read "Snomax"

P7 L22 and various other locations: "Milli-Q water" is laboratory slang, use "ultrapure water" instead

P8, L19: "flow" should read "flow rate"

P8, L19: for the flow rate reported in slpm, reference temperature and pressure need to be given

P8, L21: remove "." from "conducted.:"

P11 L6 "latter" should read "later"

P12 L5 "carbonatitie" should read "carbonatite"

P12 L6 introduce abbreviation "REE" (e.g., in the previous sentence)

P13 L23 "If a silicate components were..." – remove superfluous "a"

P16 L2 "This represents and upper limit" should read "an upper limit"

List of references: for several references, page numbers are given as "n/a-n/a", e.g. P23 L16, L24, L27; P25 L1. On P23, L30: ">" should read ">".

Table 2: In the table caption, the wording "biological filter" is unclear, please be more precise. Also not "negative particles" are "sampled", but negative ion mass spectra are acquired. For Argentina and China, "approximate" could be omitted from column 3 (as this is already clear from the column header).

References:

Després, V. R., Huffman, J. A., Burrows, S. M., Hoose, C., Safatov, A. S., Buryak, G., Fröhlich-Nowoisky, J., Elbert, W., Andreae, M. O., Pöschl, U., and Jaenicke, R.: Primary biological aerosol particles in the atmosphere: a review, Tellus Ser. B-Chem. Phys. Meteorol., 64, 2012.

Georgakopoulos, D. G., Després, V., Fröhlich-Nowoisky, J., Psenner, R., Ariya, P. A., Pósfai, M., Ahern, H. E., Moffett, B. F., and Hill, T. C. J.: Microbiology and atmospheric processes: biological, physical and chemical characterization of aerosol particles, Biogeosciences, 6, 721-737, 2009.

---

## Referee Comment (RC2) · Anonymous Referee #2 · 13 Jan 2017

Improved identification of primary biological aerosol particles using single particle mass spectrometry

By Zawadowicz et al.

General:

This paper explores the use of phosphate and organic nitrogen markers and their ratios in order to improve the separation of biological and other phosphate-containing aerosols using single-particle mass spectrometry. Overall, this paper is well-written and should be considered for publication after major revisions are made to the manuscript.

Major Comments:

[Figure]

The major concerns that I have are that the methodology used in this manuscript is not given in enough detail, which makes evaluating this method and the error analysis performed quite challenging. The paper would greatly benefit from the addition of a table with statistics of how many particles were analyzed, how many positive spectra, negative spectra, and what ion peak thresholds were used. There is also no mention of the methods used to determine misclassifications. This manuscript needs to be rewritten to include these critical details.

My other concern is that the title and introduction imply that bioaerosols will be distinguished from dust and fly ash using this method. However, it is stated in several areas of the results and discussion that what is really distinguished are biological phosphorus and inorganic phosphorus. The abstract and title should be updated to reflect what is actually being measured in this paper.

Finally, given the prevalence of mixed biological/dust particles observed in ambient observations, a more detailed discussion of experiments used to characterize these mixtures is needed.

Specific Comments:

Abstract:

1. The authors should mention the prevalence of mixed biological and dust particles.

Introduction:

1. The introduction needs to be substantially revised. In its current form, the introduction first details ice nucleation, which was not explored in this work, then discusses other methods used to identify bioaerosols, then provides a very short introduction to single particle methods of bioaerosol detection. The introduction should be more focused on methods used to distinguish bioaerosols and dust, and focus more heavily on single particle methods.

2. The last paragraph of the introduction should be cut.

3. What is the distinction between goals 1 and 2 listed in the introduction? These two goals seem quite similar to me.

4. The first paragraph of section 4.2 and a condensed version of the first paragraph of section 4.3 both belong in your introduction.

Methods

1. More detail is needed here. A table containing the statistics of how many particles were analyzed, how many positive spectra, negative spectra, and what ion peak thresholds were used.

2. How were misidentifications determined?

3. Was the same laser fluence used for all experiments including the ambient work? This could affect ion peak ratios.

4. Can it be confirmed that your experiments with illite and spores did indeed contain internally mixed particles?

5. Was a sensitivity analysis performed to confirm that your algorithm was indeed optimized for distinguishing particle types?

6. Add lines 19-20 on page 10; lines 13-15 on page 11; lines 19-22 on page 11 here. These are details of your methods.

Results

1. Page 10, lines 9-13. It seems that positive ions can also be used to filter by particle type, as was done using other single-particle methods. The authors should comment on this.

2. Page 10, lines 19-20, why were only the organic nitrogen and phosphate peaks used to distinguish these classes of aerosols. From your mass spectra, it seems that the addition of other markers could help improve the separation between different classes

of aerosols.

3. Page 10, lines 23-27: do you have an explanation for your observed changes in the phosphate ion ratios for inorganic and biological phosphorus?

4. Page 10, line 29: how are misclassifications identified and quantified?

5. Page 12, lines 7-10: it seems that this method also relies on a Boolean type of classification and not just ion peak ratios in order to distinguish aerosol types similar to the ATOFMS methods. The authors should mention that both methods are helpful for distinguishing particle types with similar ion peaks (e.g., fly ash and soil dust in this case).

6. Page 12, lines 13-15: why is the discussion of your experiments with mixed biological and dust particles not mentioned in this section? Clearly your ambient data shows that these particle mixtures are atmospherically relevant.

Discussion

1. Page 14, lines 23-26: would you be better able to distinguish bioaerosols if you applied a similar filter (e.g., if you looked for spectra containing Ca, Na, organic carbon, organic nitrogen, and P then applied your ion peak ratio determinations?)

2. Section 4.3 belongs in the results section and should be discussed in greater detail since this particle type appeared to be the most atmospherically relevant.

Technical Comments

1. Replace "species" with "compounds". Species denotes something biological.

2. Page 9, line 28, change "contamination" to "contaminant".

3. Page 16, line 2, change "and" to "an"

Please also note the supplement to this comment:

[Figure]

http://www.atmos-chem-phys-discuss.net/acp-2016-1119/acp-2016-1119-RC2-supplement.pdf

---

## Referee Comment (RC3) · Anonymous Referee #3 · 19 Jan 2017

In the mauscript "Improved identification of primary biological aerosol particles using single particle mass spectrometry" Zawadowicz et al. present a new particle classification method for a less vulnerable identification of "biological material". In detail the phosphate and organic nitrogen peaks respectively their ratios were used as markers for the assignment of either biological phosphorus or inorganic phosphorus. This is a scientific important step and fits the scope of ACP.

The paper is well written and the data basis seems to be excellent, but the general structure of the manuscript has to be revised. I would recommend the publication of this manuscript in ACP after major revisions. I have two major concerns concerning the manuscript.

1. The main message/orientation of the manuscript is not really clear. It can be seen as a methodical manuscript, but in this way there would be a lack of methodical information and critical discussion. Therefore, the reader cannot reproduce the uncertanties and in this way the quality of the new classification procedure. Some parts of the manuscript are dealing with atmospheric applications of the new classification procedure or biological particles in general. For example the introduction is dealing with very specific atmospheric questions, where biological particles have a crucial impact as their function as nutrient source or ice nuclei. But there are nor results/ discussion to these fields in the manuscript. In my opinion focusing on a critical discussion of the new procedure and all linked uncertanties would be essential to strengthen the significance and impact of this paper.

2. Based on the missing detailed methodical discussion, I am not sure if I can follow all parts of the procedural method completely. Within the manuscript a good register of the main particular sources for phosphorus is given. As the peak intensity in SPMS is not following the mass abundance only I am not convinced that only the three discussed groups (soil dust, fly ash, biological) will show phosphorus signatures in SPMS. What is with mixed particles e.g. biological layers on soil dust or sea-salt particles? Do they show phosphorus signals? There is one of the best atmospheric SPMS data sets in the world in Cambridge and Boulder, which could be used to get some additional information about the frequency of phosphorus signals in SPMA spectra at different locations. This information would help to estimate the applicability of the new classification procedure in the field.

Single points:

One question to the tested reference materials. A variety of pollen samples are tested. These pollen will have a size of > 10 $\mu$m, while the upper limit for the SPMS is given with 2-3$\mu$m. Is debris of the original pollen measured? In this way a discussion about the kind of biological particles, which should be captured with the new classification method (size, mixing-state = only external mixed biological particles or also external

mixed biological layers) would also be a great advantage.

---

## Referee Comment (RC4) · Anonymous Referee #4 · 27 Jan 2017

This paper has some very good ideas and execution behind it. And it should be published. My main concern, like the referees, is that the journal chosen is not the best fit for what is essentially an instrumentation paper to my mind. AMT would be a better fit. I also agree with the others in that the essence is not PBAP....rather a neat method of potentially distinguishing between them in the atmosphere. Given that, I thought the somewhat extensive discussion on WIBS as a bioaerosol detection technique was not necessary (although due mention should be made). The same criticism applies to the ice nucleation material. The one major scientific matter of concern to me is the laboratory work on pollen (too large for detection) unless sub-pollen. I was not at all sure what could be characterized here as the atmospheric process associated with this is

complicated involving both humidity and, when appropriate, lightning.

---

## Author Comment (AC1) · 18 Apr 2017

The authors would like to thank all reviewers for their insightful comments which we believe helped to significantly improve this paper. When Reviewers cited each other and/or in the case of similar comments we grouped these by topic first to make the responses easier to follow.

**Journal placement/atmospheric implications**

• [T]he manuscript is focused more on the method development instead of the atmospheric application, which seems only a very minor addition to the paper and doesn't provide much new insight beyond a proof-of-concept. My feeling therefore is that the manuscript would be more suited for a journal more dedicated to the technical aspects of aerosol measurements, like Atmospheric Measurement Techniques. (Reviewer #1)

• My main concern, like the referees, is that the journal chosen is not the best fit for what is essentially an instrumentation paper to my mind. AMT would be a better fit. (Reviewer #4)

The authors agree that the single field dataset in the original manuscript could be expanded. We have added a different location and season - the 2010 CARES campaign - in order to make a stronger case. See Figure 7 and associated descriptions of the CARES dataset.

We feel strongly that ACP is the right destination for this paper because the topic is of significant interest to our community, there is significant laboratory and field data, and the results affect conclusions of many previously published papers in this area. We note in particular Pratt et al., (2009) and Creamean et al., (2013) which are widely cited studies where the conclusions hinge on the use of SPMS spectra to identify biological particles. Our paper expands upon this work and suggests the urgent need for expanded analysis.

In further defending inclusion of this paper in ACP and not AMT we note that calling this "an instrumentation paper" is not correct as it includes significant laboratory work, data analysis to elucidate a contemporary issue in aerosol studies and now multiple field data sets. One could take this argument further to say a paper centered on field or lab data with incomplete analysis would be applicable to ACP but a paper with significant data analysis would not; this is not in keeping with the ACP guidance "The main subject areas comprise atmospheric modeling, field measurements, remote sensing, and laboratory studies of gases, aerosols, clouds and precipitation... The journal scope is focused on studies with general implications for atmospheric science rather than investigations that are primarily of local or technical interest."

**PBAP vs. biological phosphorus**

• While in the abstract and most of the manuscript the impression is given primary biological aerosol particles (PBAP) could be clearly distinguished from other particles, only in the last section of the discussion (4.3) it becomes apparent that this is not the case at all: in fact, what can be distinguished with the algorithm is whether the particles contain "biological" or "inorganic" phosphorus. (Reviewer #1)

• [T]he title and introduction imply that bioaerosols will be distinguished from dust and fly ash using this method. However, it is stated in several areas of the results and discussion that what is really distinguished are biological or organic phosphorus and inorganic phosphorus. The

abstract and title should be updated to reflect what is actually being measured in this paper. (Reviewer #2)
• I also agree with the others in that the essence is not PBAP....rather a neat method of potentially distinguishing between them in the atmosphere. (Reviewer #4)

In order to address these concerns, a clear definition of "bioaerosol" as used in this paper is now in the introduction: "In this paper, "bioaerosol" is defined as primary biological aerosol particles (PBAP) (i.e. airborne whole and fragmentary bacteria, pollen and spores) and particles that contain fragments of PBAP as a part of an internal mixture." The following wording was also added in the introduction, "In this work, the presence of phosphorus in a mass spectrum is used as proxy for bioaerosol. All biological cells contain phosphorus because it is a component of nucleic acids and cell membranes. Distinguishing the specific phosphate signature of biological cells from other non-biological phosphorus is the topic of the analysis in this paper."

Uncertainties present in soil dust particles are now discussed, see the responses under "More focused discussion of uncertainties and limitations" heading in this document.

**Introduction length and detail**
• [T]he introduction is very long, and in parts reads more like a review than like a general introduction to a research paper, esp. P2, L22 to P4, L28. This also leads to an excessively long list of references. (…) On the other hand, the discussion on previous efforts in bioaerosol detection using single particle mass spectrometry, which is the most important aspect for the discussions within the manuscript, is very brief and could be expanded (P5 L9-16). (Reviewer #1)
• The introduction needs to be substantially revised. In its current form, the introduction first details ice nucleation, which was not explored in this work, then discusses other methods used to identify bioaerosols, then provides a very short introduction to single particle methods of bioaerosol detection. The introduction should be more focused on methods used to distinguish bioaerosols and dust, and focus more heavily on single particle methods. (Reviewer #2)
• [T]he introduction is dealing with very specific atmospheric questions, where biological particles have a crucial impact as their function as nutrient source or ice nuclei. But there are nor results/ discussion to these fields in the manuscript. (Reviewer #3)
• I thought the somewhat extensive discussion on WIBS as a bioaerosol detection technique was not necessary (although due mention should be made). The same criticism applies to the ice nucleation material. (Reviewer #4)

The introduction was revised and shortened per these comments. In particular, ice nucleation material was shortened (a short overview was retained for context), the paragraph about bioaerosols in cloud water has been cut, and material about previous measurements of bioaerosol (including WIBS) is now only a short overview.

The discussion of previous efforts to detect bioaerosol with SPMS was expanded. Per the last point a clarification about bio/dust mixtures has been added: "Particles that contain

phosphates, organic nitrates and silicates have been historically always classified as mixtures of bioaerosol and dust (Creamean et al., 2013)." This section is difficult to expand further, as those methods are very poorly characterized in previous literature (this paper aims to improve on that).

• The last paragraph of the introduction should be cut. (Reviewer #2)

We decided to keep a shortened version of the paragraph in order to maintain motivation for using phosphorus in this analysis (this could not be eliminated): "Phosphorus was chosen as the focus of this paper because of its abundance in spectra of bioaerosol, but also because it does not undergo gas-phase partitioning in the atmosphere (Mahowald et al., 2008). Therefore, the presence of phosphorus on a particle can often constrain its source and only the classes of particles that are most likely to contain phosphorus are examined here. Emission estimates qualitatively agree that mineral dust, combustion products, and biological particles constitute the principal phosphate emission sources."

• What is the distinction between goals 1 and 2 listed in the introduction? These two goals seem quite similar to me. (Reviewer #2)

The paper has been reworded to avoid the distinction: "This work examines the prevalence of these ions in the context spectra collected with PALMS" and "The goal of this paper is to develop a method that can differentiate PALMS bioaerosol spectra from spectra of dust and combustion by-products."

• The first paragraph of section 4.2 and a condensed version of the first paragraph of section 4.3 both belong in your introduction. (Reviewer #2)

Change made: The following has been added to the introduction: "In atmospheric particles, the composition can be mixed, containing some phosphate from inorganic sources, such as calcium phosphate, and some phosphate from microbes. For instance, soils can contain minerals, live microbes, and biogenic matter at all stages of decomposition. Therefore, classifying soil-derived particles with a binary biological/non-biological classifier has uncertainties. These uncertainties are quantified here for soils using soil samples collected in various locations." And "Biological aerosols have been studied with SPMS, in particular the Aerosol Time of Flight Mass Spectrometer (ATOFMS; Cahill et al., 2015; Creamean et al., 2013; Fergenson et al., 2004; Pratt et al., 2009). A property of SPMS bioaerosol spectra that has been exploited for their detection is the presence of phosphate ($PO^-$, $PO_2^-$, $PO_3^-$) and organic nitrogen ions ($CN^-$, $CNO^-$) (Cahill et al., 2015; Fergenson et al., 2004). Those ions have also been shown to be present in non-biological particles with the same instrument, however, such as vehicular exhaust (Sodeman et al., 2005) and soil dust (Silva et al., 2000). Particles that contain phosphates, organic nitrates and silicates have been historically always classified as mixtures of bioaerosol and dust (Creamean et al., 2013). This work examines the prevalence of these ions in the context spectra collected with PALMS."

**More streamlined discussion of samples used/table of particle numbers**

• In order to get an idea of the underlying statistics, the general information on how many mass spectra (positive/negative) were available for the different samples needs to be included somewhere (in the experimental or the results section), also, how many mass spectra were acquired in the field campaign. (Reviewer #1)

• Could Section 2.2 (Test samples) be streamlined a bit to be more concise? It would be good to have a table with an overview of the sampled materials; maybe some of the detailed information could go into such a table as well, to make the section easier to read and to provide a better overview for the reader. (Reviewer #1)

• P9, L3: "a portion of the data" – how many spectra? Give at least the order of magnitude. The same for the "remaining data" (P9 L5). Also specify which sample types were included for the training: on P11 L5 it is mentioned that soil data were not used. Were all other lab samples used? Please be more specific. (Reviewer #1)

• P10 L19 to L27: which bioaerosol materials were used for this? In Fig. 4 only 5 of the 8 tested materials are shown. Were the others used as well, showing similar behavior, and only were omitted for clarity? Please state clearly which materials were used for the development and if some were left out, why. – On P11 L4 it is stated that soil dust is left out from the training set because some biological material might be contained in the particles, however, in P10 L24 it is used within the class of inorganic phosphorus. If biological material is indeed present within soil dust, this does not make sense. This seems like a contradiction. (Reviewer #1)

• The paper would greatly benefit from the addition of a table with statistics of how many particles were analyzed, how many positive spectra, negative spectra, and what ion peak thresholds were used. (Reviewer #2)

Several steps were taken to address the confusion regarding which samples formed the training dataset and which did not:

1. Section 2.2 was clarified and re-structured. The section is now split into two sub-sections, "2.2.1 Training dataset" and "2.2.2 Test dataset" to make sure that confusion does not occur. All detail about sample processing was retained, as the new organization should make it easier to follow.

2. Added clarifying remarks in section 3 and Figure 4 caption, "Soil dusts are shown in Figure 4, even though they are not used as training aerosol; their histogram shows a broad distribution with a tail extending into $PO_3^-/PO_2^-$ > 2 region, indicating a mixed inorganic/biological composition. In comparison, fertilized soil dusts show a similar distribution to apatite ($PO_3^-/PO_2^-$ < 4) due to presence of inorganic fertilizer, which is calcium phosphate" and "Note that soil dusts were not used as part of the training dataset and that not all training aerosols are shown here for clarity."

3. Added Table 2 detailing all aerosol samples used and their statistics. The table should also make it clear which aerosol made it into the training set.

**More focused discussion of uncertainties and limitations**

• Section 4.2: This is an important discussion, and I would have hoped to find a similarly critical discussion of the newly developed algorithm in this manuscript, as well. There are some basic approaches to such a discussion scattered throughout the manuscript, but this should be addressed much more clearly and explicitly. By the title of Section 4.1 it is suggested that this discussion is provided in that section, but in fact the presented discussion on uncertainties and limitations of the newly developed algorithm in that section is very limited and should be much more thorough. (Reviewer #1)

• In my opinion focusing on a critical discussion of the new procedure and all linked uncertanties would be essential to strengthen the significance and impact of this paper. (Reviewer #3)

The Discussion section has been restructured. Material noted as irrelevant to the discussion of uncertainties has been taken out of section 4.1 and shortened and placed at the beginning of the Discussion. Moreover, section 4.1 was shortened and moved to section 4.3.

Additional quantification of uncertainty has been added to the SVM classification approach: "For every observation, a distance from the SVM boundary can be calculated (otherwise known as score). Those distances can then be converted to probability of correct identification. An optimized function to convert scores to probabilities was found by 10-fold cross-validation (Platt, 1999). Because in this experiment the classes are not perfectly separable, the conversion function is a sigmoid. Posterior probabilities near 0 and 1 indicate high-confidence identification. An uncertainty boundary was defined between 0.2 and 0.8. This boundary is shown in Figure 5. Points that lie in this boundary are marked as low confidence assignments. Those correspond to shaded areas in Figures 6 and 7." Changes have been made to figures 5 and 6 accordingly.

Further discussion of the uncertainty bound was included in section 4.3 : "The basic classifier presented in this paper is binary: all phosphate- and organic nitrogen-containing particles are classified either as biological or inorganic. However, spectra whose $PO_3^-/PO_2^-$ and $CN^-/CNO^-$ ratios are very close to the SVM boundary have more uncertain assignments than those whose $PO_3^-/PO_2^-$ and $CN^-/CNO^-$ ratios fall far away from the boundary. In order to provide an additional measure of classification uncertainty, a probability bound was defined as shown in Figure 5. According to this definition, 96% of particles in the training dataset were classified with high-confidence (Figure 5). In the FIN03 and CARES field datasets, 79% of phosphate-containing particles were classified with high confidence. The low-confidence assignments are shown on Figures 6A and 7A with shaded areas."

Section 4.3 (previously section 4.1) now includes a discussion of uncertainties related to dust/biological mixtures and soil dusts: "Because soil dusts are a special category, where lines between biological and inorganic phosphorus sources can be blurry because of ongoing chemical transformations, they have higher classification uncertainties than other types of

phosphate-containing aerosols. In the field data, dust/biological mixtures (defined as particles classified as biological with silicate features) are overrepresented in the low-confidence assignments. Dust/biological mixtures constitute 26% (CARES) - 46% (FIN03) of high-confidence assignments and 64% (CARES) - 68% (FIN03) of low-confidence assignments. Moreover, only 75% of phosphate-containing soil dust particles were classified with high confidence. However, in simple two-component internal mixtures of dust and biological fragments (Figure 10) phosphate features can be identified as biological with high confidence (98%)."

**Discussion of misclassification rate**
• 1.) What kind of misidentification exactly is contained within this value? Both false positives (mineral dust wrongly assigned to "biological phosphorus") and false negatives (biological material not assigned to "biological phosphorus") cause an uncertainty in the determined fraction of particles containing biological phosphorus. (Reviewer #1)
• 2.) The method used to calculate the "misidentification rate" should be clearly stated. Depending on the method, the number of particles within the data sets of the different test aerosols might bias the determined uncertainty, so this should be made clear to the reader. (Reviewer #1)
• 3.) It is stated that removing ragweed pollen from the training set leads to a smaller misidentification rate. Was ragweed pollen only removed from the training set, or also from the "testing" set? I guess the former, since the latter would give a wrong impression, but this needs to be stated clearly. (Reviewer #1)
• 4.) Which particles were tested for determining this misidentification rate? The pure biological / pure mineral dust / fly ash particles? What about the processed mineral dust? (Reviewer #1)
• How were misidentifications determined?   (Reviewer #2)
• Page 10, line 29: how are misclassifications identified and quantified? (Reviewer #2)

In order to avoid confusion the term "accuracy" was used throughout the paper defined as: "Accuracy here is defined as percentage of correctly classified particles in the training set once the optimized boundary is found."

Context for the SVM method was provided: "The SVM algorithm was used here to optimize boundaries between clusters. To do this, the algorithm needs a training dataset, where the classes are known *a priori*. In this paper, the training dataset is defined in Table 2. Once an optimized boundary is drawn, some of the training data can still fall on the incorrect side of the boundary, when the clusters are not perfectly separable." Note that a table detailing the training set contents was added as Table 2.

We believe that the clarification above, together with Table 2 answers the questions about which particles were used to calculate the accuracy and removing the ragweed pollen from the training set.

**Discussion of dust/biological mixtures**

• - It should be discussed in Sect. 3 / Sect. 4.1 what effect has (…)
2) mixing mineral dust / biological material. This is discussed only in Sect. 4.3, but is an important consideration when assessing the uncertainty and limitations of the method. The discussion in Sect. 4.3 reveals several limitations which need to be discussed within this context: "At this time, we are not able to delineate between primary biological and biogenic or simply complex organic (such as humic acids) material." (P15 L30-31) This means that with the presented method, not PBAP can be determined, but whether phosphorus present in any particle is of "biological" or "inorganic" nature. The former hints at the presence of biological material, but, as also evident from the discussion on P16 L1-14, it is not possible to determine whether this biological material is part of PBAP or from an internal mixture of e.g. mineral dust and biological material, so the information retrieved remains limited, which needs to be clearly stated and discussed. (Reviewer #1)

The definition of "bioaerosol" is stated in the introduction: "In this paper, "bioaerosol" is defined as primary biological aerosol particles (PBAP) (i.e. airborne whole and fragmentary bacteria, pollen and spores) and particles that contain fragments of PBAP as a part of an internal mixture." Under this definition, internal mixtures of biological material and dust also qualify as bioaerosol.

Uncertainties related to soil dusts are now discussed in more detail, described in section 4.3. See also our comments under "More focused discussion of uncertainties and limitations" heading in this document.

• - If I understand correctly, only an internal mixture of biological material with a type of mineral dust not showing any signatures of inorganic phosphorus (illite) was tested. But what happens if mineral dust showing mass spectral signatures of inorganic phosphorus (like apatite) is internally mixed with biological material? If indeed in an ambient dataset up to 56% of all particles identified as containing biological phosphorus also contained silicate markers (P16 L1), this does not seem to be an unlikely case and needs to be addressed. (Reviewer #1)

We agree with the reviewer that an internally mixed inorganic and biological phosphorous particle is a possible atmospheric state. This is now stated in section 4.3: "The low-confidence assignments in field datasets can be related to chemical processing of particles (either at the source like in soils or during transport) or to internal mixing of biological and inorganic phosphate." To be clear, we did not test this type in the laboratory.

• [G]iven the prevalence of mixed biological/dust particles observed in ambient observations, a more detailed discussion of experiments used to characterize these mixtures is needed. (Reviewer #2)
• Section 4.3 belongs in the results section and should be discussed in greater detail since  this particle type appeared to be the most atmospherically relevant.  (Reviewer #2)

• Page 12, lines 13-15: why is the discussion of your experiments with mixed biological and dust particles not mentioned in this section? Clearly your ambient data shows that these particle mixtures are atmospherically relevant. (Reviewer #2)

More detail has now been included in section 4.3 (see responses above and comments under "More focused discussion of uncertainties and limitations" heading in this document). We tried to confine this discussion to one section to keep it focused because of its importance.

• The authors should mention the prevalence of mixed biological and dust particles [in the abstract]. (Reviewer #2)

The following was added, "In addition, 36% - 56% of particles identified as biological also contained spectral features consistent with mineral dust, suggesting internal dust/biological mixtures."

• Can it be confirmed that your experiments with illite and spores did indeed contain internally mixed particles? (Reviewer #2)

Revised wording: "Internal mixtures of biological and mineral components were generated in the laboratory in order to investigate this; an exemplary spectrum of such particle is shown in Figure 10. The spectrum contains alumino-silicate markers consistent with mineral dust together with phosphate markers that, in this case, come from the biological material. In spectra of pure illite, no phosphate markers are present."

**Discussion of processing of mineral dust**
• - It should be discussed in Sect. 3 / Sect. 4.1 what effect has
1) processing of the mineral dust (which, as stated on P10 L17, causes CN- and CNO- to "appear and/or intensify", so might have an influence on the classification) (Reviewer #1)

A clarification was added to section 3, "Processed mineral dust had a smaller impact on the accuracy: removing it from the training dataset increased the accuracy to 97.5%."

**Discussion of thresholds**
• If only some of the spectra were used for the analysis, their number (and criteria for their selection) needs to be stated. (Reviewer #1)
• [I]n Sect. 3, it is stated that for the ambient data, a threshold was used to determine mass spectra containing phosphorus in a first step. Was something similar performed for the lab data (also for the training), or were all mass spectra used? (Reviewer #1)
• P13 L26 and following: What thresholds were used in order to determine whether the different marker ion signals were present or absent? Was it tried to improve the performance of the algorithm by adjusting these thresholds? (Reviewer #1)

We have added in section 3, "The only requirement for this analysis was that each spectrum used in the training set contains both phosphate and organic nitrogen (otherwise the ratios used here become undefined). This was ensured by selecting spectra, where $PO_2^- > 0.001$ and $CNO^- > 0.001$. Nearly all biological spectra in the training set satisfied this criterion (Table 2)."

Also added the following in section 4.1 (formerly section 4.2), "Note that previous literature does not provide information on the thresholds used to determine presence or absence of ions in analysis of ATOFMS spectra. Furthermore, because of hardware differences, detection limits of PALMS and ATOFMS are very different. This analysis focuses on PALMS and the threshold for "presence" was chosen as 0.001, which was observed to be the detection limit for $CN^-$, $CNO^-$ and $PO_3^-$ in the laboratory aerosol database used here."

**Pollen**
• One question to the tested reference materials. A variety of pollen samples are tested. These pollen will have a size of > 10 μm, while the upper limit for the SPMS is given with 2-3μm. Is debris of the original pollen measured? In this way a discussion about the kind of biological particles, which should be captured with the new classification method (size, mixing-state = only external mixed biological particles or also external mixed biological layers) would also be a great advantage. (Reviewer #3)
• The one major scientific matter of concern to me is the laboratory work on pollen (too large for detection) unless sub-pollen. I was not at all sure what could be characterized here as the atmospheric process associated with this is complicated involving both humidity and, when appropriate, lightning. (Reviewer #4)

Yes, we are measuring pollen fragments. This was covered in the experimental section, "Pollen grains were too large (18.9 – 37.9 μm according to manufacturer's specification) to sample with PALMS. They were suspended in ultrapure water (18.2 MΩ cm, Millipore, Bedford, MA) and the suspensions were sonicated in ultrasonic bath for ~30 minutes to break up the grains."

**All other concerns raised**
• How were peak intensities determined for the various ratios (CN-/CNO- etc)? Integrated peak area? This should be stated in the methods section. (Reviewer #1)

Added to the end of section 2.1: "Raw PALMS spectra are processed using custom IDL software. Mass peak intensities used in this paper refer to integrated peak areas normalized by the total ion current."

• P6, L3-9 (first paragraph): this paragraph is not related to the section (2.1: PALMS), but a general introduction. It should go either as a general remark in the experimental section before Sect. 2.1, or be reworked as a last paragraph into the introduction. (Reviewer #1)

This paragraph has now been revised and moved before section 2.1.

• P6, L18: "...a unipolar reflectron time of flight mass spectrometer was used..." – It should be clearly stated here that the PALMS acquires for each single particle either a negative or a positive mass spectrum, but not both simultaneously. How long were the sampling times in positive / negative mass spectra mode (e.g., switching every minute, every 15 min, every hour)? (Reviewer #1)

The following was added to 2.1: "PALMS acquires spectra in either positive or negative polarity, but not simultaneously. For field datasets presented in this paper, sampling polarity was switched every 5 minutes for FIN03 and every 30 minutes for CARES."

• P7 L25: ". . .further dissolved in ~5ml of Milli-Q water. . ." – this information is not necessary since the concentration of the original solution is not known, anyway. (Reviewer #1)

The estimated volume was removed: "…further dissolved in ultrapure water,"

• P7, L30-31: "No processing-related changes to chemistry were found." – This sentence should be clarified, e.g., ". . .were found in the mass spectra sampled with the PALMS". (Reviewer #1)

Clarified as follows: "Examination of PALMS spectra revealed no changes in chemistry resulting from different processing methods."

• P8 L4 "to aerosolize a solution of illite NX and F. solani spores" – I guess this should read "suspension" instead of "solution"? Was this suspension sonicated as well? (Reviewer #1)

Changed to "suspension" instead of "solution" and clarified: "A second disposable medical nebulizer was then used to aerosolize a suspension of illite NX and *F. solani* spore fragments."

This suspension was not sonicated, but the spores mixed with illite NX were actually fragments prepared by sonication, as described above.

• P8 L21: "for 0.1 mL experiments" is unclear. Rather something like "For experiments using 0.1 mL of nitric acid"? (Reviewer #1)

Changed to "…for experiments using 0.1 mL of nitric acid, the entire volume of $HNO_3$ evaporated…"

• P9, L18: Please give the start / end dates of the measurement period. (Reviewer #1)

Added the following to the end of section 2.4: "The measurements were carried out between September 14, 2015 and September 27, 2015."

• It is not clear to me why parts of the results can be found in the results section, and other parts (like the soil and internal mixtures, Sect. 4.3) in the discussion section. Maybe a single

"results and discussion" section would be more appropriate, and could also avoid some repetitions. (Reviewer #1)

Both sections were screened for repetitions, which were edited out when applicable (see track changes version).

• P10 L26 "Processing of apatite with nitric acid tends to shift the PO3-/PO2- ratio to larger values, increasing the disparity from the bioaerosols." – This is not clear to me. If inorganic material usually shows lower PO3-/PO2- ratios than bioaerosol, shouldn't this read "decreasing the disparity"? (Reviewer #1)

The Reviewer is correct, it should be "decreasing": "Processing of apatite with nitric acid tends to shift the $PO_3^-/PO_2^-$ ratio to larger values, decreasing the disparity from the bioaerosols."

• P11 L6: "(classification with the SVM algorithm is discussed later)": maybe this could be reworded for clarity, as at first reading it seems to mean that the SVM algorithm itself will be discussed later, not the results of applying it on the soil samples. (Reviewer #1)

Clarified as follows: "(classification of soil dusts with the SVM algorithm is discussed later)"

• P13 L10-12: "Particles with positive spectra showing the characteristics of monazite. . .provides evidence of the origin of the inorganic phosphate particles." (and a similar statement on P12, L9): Since PALMS does not simultaneously provide the positive and negative ion mass spectrum of a single particle, this is not "evidence", but rather "suggests" this type of mineral dust particles as a likely origin. Please reword. (Reviewer #1)

The Reviewer is correct. Clarified as follows: "Although negative spectra of apatite and monazite cannot be definitively differentiated from fly ash or soil dust spectra, positive spectra acquired during FIN03 additionally suggest that monazite-type material was present."

And later, "Particles with positive spectra showing the characteristics of monazite coupled to back trajectories over source areas suggests the origin of the inorganic phosphate particles."

• P15 L26/27: "the numbers of biological particles fall within these estimates". The cited estimates refer to inorganic / biological phosphorus mass ratio, while the information provided by PALMS is the number ratio. Such a comparison would only be valid if it can be assumed that all types of particles contain the same amount of phosphorus, which does not seem very likely. This needs to be discussed more carefully. A similar comment applies to P16, L28-31. (Reviewer #1)

This wording was removed to avoid confusion, "The biological PALMS filter was applied to several soil dust samples (Table 2). As would be expected, soils collected in areas with less vegetation exhibit smaller biological contributions."

• Figs 1-3, Fig. 9: The mass spectra might be easier to grasp if integrated stick spectra were shown instead of the raw mass spectra. (Reviewer #1)

We disagree in this case; spectra are shown in a manner consistent with previous PALMS publications. We believe a change in this paper would have made future comparisons to prior or future by readers difficult.

• Fig. 7: "In all other aerosol classes the green bar denotes a level of misidentification." – This only applies to "Apatite + Monazite" and "Fly ash". Both "agricultural soil dust" and "Soil dusts" are expected to contain some (unknown) amount of biological material, so the performance of the algorithm cannot be validated on these samples. (Reviewer #1)

This was reworded, "In all other aerosol classes the green bar denotes a typical level of misidentification."

• Various locations, e.g., P6 L6, P12 L23, P12 L28: "phosphorous" should read "phosphorus" (Reviewer #1)

Corrected.

• P7 L8: "Snowmax" should read "Snomax" (Reviewer #1)

Corrected.

• P7 L22 and various other locations: "Milli-Q water" is laboratory slang, use "ultrapure water" instead (Reviewer #1)

Corrected.

• P8, L19: "flow" should read "flow rate" (Reviewer #1)

Corrected.

• P8, L19: for the flow rate reported in slpm, reference temperature and pressure need to be given (Reviewer #1)

Corrected, "…flow rate of 0.44 slpm (STP: 0°C, 1 atm)…"

• P8, L21: remove "." from "conducted.:"  P11 L6 "latter" should read "later"  P12 L5 "carbonatitie" should read "carbonatite" (Reviewer #1)

Corrected.

• P12 L6 introduce abbreviation "REE" (e.g., in the previous sentence) (Reviewer #1)

Changed to "As examples, on 09/27 the back trajectory intersects the vicinity of an active rare earth element (REE) mine…"

• P13 L23 "If a silicate components were. . ." – remove superfluous "a" (Reviewer #1)

Corrected.

• P16 L2 "This represents and upper limit" should read "an upper limit" (Reviewer #1)

Corrected.

• List of references: for several references, page numbers are given as "n/a-n/a", e.g. P23 L16, L24, L27; P25 L1. On P23, L30: ">" should read ">".(Reviewer #1)

Corrected.

• Table 2: In the table caption, the wording "biological filter" is unclear, please be more precise. Also not "negative particles" are "sampled", but negative ion mass spectra are acquired. For Argentina and China, "approximate" could be omitted from column 3 (as this is already clear from the column header). (Reviewer #1)

Table 2 caption changed to read, "Soil dust samples used in this work. The last column shows the results of analysis with the SVM classifier developed here as a percentage of negative spectra acquired."

"Approximate" left off for Argentina and China.

• Was the same laser fluence used for all experiments including the ambient work? This could affect ion peak ratios.  (Reviewer #2)

The following statement was added to section 4.3, "Because the field studies were performed during very different time periods, it was difficult to control for a constant excimer laser fluence. However, laser fluence was similar for all laboratory samples acquired (3-5 mJ pulse energy). This is a possible source of uncertainty, as fragmentation patterns can differ depending on pulse energy."

• Was a sensitivity analysis performed to confirm that your algorithm was indeed optimized for distinguishing particle types?  (Reviewer #2)

The following wording was added to the results section, "Those spectral peaks were used for several reasons: (1) they are clearly visible in all biological spectra that were acquired as a part of this study (Figure 1), (2) they were used to distinguish bioaerosol from other species in previous studies (Creamean et al., 2013; Pratt et al., 2009) and (3) sources of phosphorus on aerosol particles are well-defined and documented in the literature (Mahowald et al., 2008)." For these reasons we did not perform a sensitivity analysis.

- Add lines 19-20 on page 10; lines 13-15 on page 11; lines 19-22 on page 11 here. These are details of your methods. (Reviewer #2)

In the revised manuscript we believe details of thresholds and SVM development in the results/discussion section was found to streamline the discussion.

- Page 10, lines 9-13. It seems that positive ions can also be used to filter by particle type, as was done using other single-particle methods. The author should comment on this. (Reviewer #2)

In this work, we use negative markers to compare to previous methods of distinguishing bioaerosol from other particle classes (the Boolean phosphate and organic nitrogen markers). Those markers are strongly visible only in negative spectra (see Figure 1). See the following added explanation, "Those spectral peaks were used for several reasons: (1) they are clearly visible in all biological spectra that were acquired as a part of this study (Figure 1), (2) they were used to distinguish bioaerosol from other species in previous studies (Creamean et al., 2013; Pratt et al., 2009) and (3) sources of phosphorus on aerosol particles are well-defined and documented in the literature (Mahowald et al., 2008)."

- Page 10, lines 19-20, why were only the organic nitrogen and phosphate peaks used to distinguish these classes of aerosols. From your mass spectra, it seems that the addition of other markers could help improve the separation between different classes of aerosols. (Reviewer #2)

The following wording was added : "Those spectral peaks were used for several reasons: (1) they are clearly visible in all biological spectra that were acquired as a part of this study (Figure 1), (2) they were used to distinguish bioaerosol from other species in previous studies (Creamean et al., 2013; Pratt et al., 2009) and (3) sources of phosphorus on aerosol particles are well-defined and documented in the literature (Mahowald et al., 2008)." Also note that markers in positive spectra cannot be used concurrently with markers in negative spectra because PALMS does not acquire positive and negative spectra simultaneously (now clarified, per Reviewer #1 comment above).

- Page 10, lines 23-27: do you have an explanation for your observed changes in the phosphate ion ratios for inorganic and biological phosphorus? (Reviewer #2)

Different chemical forms of phosphorus in those different particle classes are a possible explanation, now stated in section 4.1: "In apatite and monazite, phosphorus occurs as calcium phosphate. In biological particles, phosphorus occurs mostly in phospholipid bilayers and nucleic acids. In these experiments, the $PO_3^-/PO_2^-$ ratio of those two forms is different (Figure 4A). The agricultural soils considered here cluster with the minerals and fly ash and we assume the phosphorus is due to the use of inorganic fertilizer, which is derived from calcium phosphate (Koppelaar and Weikard, 2013). Fly ash aerosol clusters similarly to apatite and monazite but with a wider distribution; this is likely because the chemical from of phosphorus in fly ash is different than in the minerals. Phosphorus present in coal is volatilized and then condenses into different forms during the combustion process (Wang et al., 2014)."

• Page 12, lines 7-10: it seems that this method also relies on a Boolean type of classification and not just ion peak ratios in order to distinguish aerosol types similar to the ATOFMS methods. The authors should mention that both methods are helpful for distinguishing particle types with similar ion peaks (e.g., fly ash and soil dust in this case). (Reviewer #2)

Added the following to section 4.1, "Such "Boolean" criteria for particle identification, can be helpful in distinguishing aerosol types when the signatures are unique to one particle type."

• Page 14, lines 23-26: would you be better able to distinguish bioaerosols if you applied a similar filter (e.g., if you looked for spectra containing Ca, Na, organic carbon, organic nitrogen, and P then applied your ion peak ratio determinations?) (Reviewer #2)

As clarified above, only negative spectra were used here for consistency with previous data analyses.

• Replace "species" with "compounds". Species denotes something biological. (Reviewer #2)

"Species" replaced with "types of particles" in Figure 4 caption, "Delineation between the clusters at a $PO_3^-/PO_2^-$ ratio of 3 results in a 70-80% classification accuracy depending on the types of particles considered."

"Species" replaced by "compounds" and "ions" in the Results section: "These particles contain both organic and inorganic compounds. Because they are easy to ionize, the inorganic ions sodium and potassium stand out in the positive spectra despite their minor fraction by mass."

• Page 9, line 28, change "contamination" to "contaminant". (Reviewer #2)

Corrected.

• Page 16, line 2, change "and" to "an" (Reviewer #2)

Corrected.

• As the peak intensity in SPMS is not following the mass abundance only I am not convinced that only the three discussed groups (soil dust, fly ash, biological) will show phosphorus signatures in SPMS. What is with mixed particles e.g. biological layers on soil dust or sea-salt particles? Do they show phosphorus signals? (Reviewer #3)

We revised the introduction: "Emission estimates qualitatively agree that mineral dust, combustion products, and biological particles constitute the principal phosphate emission sources. (…) In this work, calcium phosphate-rich minerals (apatite and monazite) and fly ash are chosen to represent dust and industrial combustion particle classes, respectively."

Note that the selection of those classes is based on the work of Mahowald, et al. (2008), which does not show sea salt as an important source of atmospheric phosphorus.

Mixed particles (mineral dust with biological fragments) were investigated in this study using both laboratory particle mixtures and natural soil samples. Those tend to be frequent in field datasets (as a possible soil dust influence). We added some uncertainty analysis regarding those particles, as described in the responses above.

**References**

Cahill, J. F., Darlington, T. K., Fitzgerald, C., Schoepp, N. G., Beld, J., Burkart, M. D. and Prather, K. A.: Online Analysis of Single Cyanobacteria and Algae Cells under Nitrogen-Limited Conditions Using Aerosol Time-of-Flight Mass Spectrometry, Anal. Chem., 87(16), 8039–8046, doi:10.1021/acs.analchem.5b02326, 2015.

Creamean, J. M., Suski, K. J., Rosenfeld, D., Cazorla, A., DeMott, P. J., Sullivan, R. C., White, A. B., Ralph, F. M., Minnis, P., Comstock, J. M., Tomlinson, J. M. and Prather, K. A.: Dust and Biological Aerosols from the Sahara and Asia Influence Precipitation in the Western U.S., Science, 339(6127), 1572–1578, doi:10.1126/science.1227279, 2013.

Fergenson, D. P., Pitesky, M. E., Tobias, H. J., Steele, P. T., Czerwieniec, G. A., Russell, S. C., Lebrilla, C. B., Horn, J. M., Coffee, K. R., Srivastava, A., Pillai, S. P., Shih, M.-T. P., Hall, H. L., Ramponi, A. J., Chang, J. T., Langlois, R. G., Estacio, P. L., Hadley, R. T., Frank, M. and Gard, E. E.: Reagentless Detection and Classification of Individual Bioaerosol Particles in Seconds, Anal. Chem., 76(2), 373–378, doi:10.1021/ac034467e, 2004.

Koppelaar, R. H. E. M. and Weikard, H. P.: Assessing phosphate rock depletion and phosphorus recycling options, Glob. Environ. Chang., 23(6), 1454–1466, doi:10.1016/j.gloenvcha.2013.09.002, 2013.

Mahowald, N., Jickells, T. D., Baker, A. R., Artaxo, P., Benitez-Nelson, C. R., Bergametti, G., Bond, T. C., Chen, Y., Cohen, D. D., Herut, B., Kubilay, N., Losno, R., Luo, C., Maenhaut, W., McGee, K. A., Okin, G. S., Siefert, R. L. and Tsukuda, S.: Global distribution of atmospheric phosphorus

sources, concentrations and deposition rates, and anthropogenic impacts, Global Biogeochem. Cycles, 22(4), GB4026, doi:10.1029/2008GB003240, 2008.

Platt, J. C.: Probabilistic outputs for support vector machines and comparisons to regularized likelihood methods, in Advances in Large Margin Classifiers, pp. 61–74, MIT Press., 1999.
Pratt, K. A., DeMott, P. J., French, J. R., Wang, Z., Westphal, D. L., Heymsfield, A. J., Twohy, C. H., Prenni, A. J. and Prather, K. A.: In situ detection of biological particles in cloud ice-crystals, Nat. Geosci., 2, 398–401, 2009.

Silva, P. J., Carlin, R. A. and Prather, K. A.: Single particle analysis of suspended soil dust from Southern California, Atmos. Environ., 34(11), 1811–1820, doi:10.1016/S1352-2310(99)00338-6, 2000.

Sodeman, D. A., Toner, S. M. and Prather, K. A.: Determination of Single Particle Mass Spectral Signatures from Light-Duty Vehicle Emissions, Environ. Sci. Technol., 39(12), 4569–4580, doi:10.1021/es0489947, 2005.

Wang, R., Balkanski, Y., Boucher, O., Ciais, P., Peñuelas, J. and Tao, S.: Significant contribution of combustion-related emissions to the atmospheric phosphorus budget, Nat. Geosci., 8(1), 48–54, doi:10.1038/ngeo2324, 2014.